# GUARD: General Unsupervised Adversarial Robust Defense for Deep Multi-View Clustering via Information Bottleneck

## Abstract

The integrity of Deep Multi-View Clustering (DMVC) is fundamentally challenged by adversarial attacks, which corrupt the learning process by injecting a malicious, task-misaligned informational signal. Existing adversarial defense methods for DMVC are model-specific, non-transferable, and limited to complete multi-view scenarios. To address this, we introduce Multi-view Adversarial Purification (MAP), a novel defense paradigm that reframes unsupervised purification as a principled, information-theoretic problem of signal separation. We present GUARD, the first framework to operationalize the MAP paradigm, which instantiates the principles of the Multi-View Information Bottleneck. GUARD is designed to satisfy a dual objective: 1) it maximizes informational sufficiency with respect to the benign data, ensuring the preservation of all task-relevant information; and 2) it enforces purity against the adversarial signal by creating a bottleneck to discard it. Crucially, GUARD achieves this duality not with an explicit penalty term, but through a unique self-supervisory design where the information bottleneck emerges as a property of the optimization dynamics. Extensive experiments validate that our model-agnostic and unsupervised framework effectively purifies adversarial data, significantly enhancing the robustness of a wide range of DMVC models.

## 1 Introduction

Multi-view data, representing the same object from different perspectives, has attracted significant attention in the field of machine learning (Geng et al., 2024; Yu et al., 2025). Clustering, a fundamental problem in unsupervised learning, has been extensively studied in the context of multi-view data (Fang et al., 2023; Ren et al., 2024; Zhou et al., 2024b). DMVC methods have achieved significant success by harnessing the complementary and consistent information across various views (Zhou et al., 2024a), even in real-world scenarios where data may be incomplete (Yang et al., 2022; Wen et al., 2022; Zhang et al., 2024). However, the very complexity that allows these models to succeed also creates a critical vulnerability. The integrity of DMVC is fundamentally challenged by adversarial attacks, which can corrupt the clustering process and lead to unreliable outcomes, making the development of effective defense mechanisms a pressing priority (Huang et al., 2024).

The challenge of defending DMVC is a nascent area, with research to date focused on Adversarial Training (AT). The pioneering work in this space, AR-DMVC (Huang et al., 2024), enhances robustness by retraining the clustering model on adversarial examples. However, this AT-based methodology has significant drawbacks: it can degrade performance on clean data, and the resulting defense is model-specific, non-transferable, and limited to complete multi-view scenarios. The limitations of this approach motivate exploring alternative paradigms from the broader field of adversarial defense. This raises a key research question: *Can we design a model-agnostic adversarial defense method that is explicitly aware of the multi-view data structure?*

To answer this question, in this paper, we propose the first Multi-view Adversarial Purification (MAP) framework tailored to multi-view clustering. Unlike prior AP methods developed for single-view tasks (Han et al., 2025) that operate on each input independently, our approach is explicitly designed to preserve cross-view consistency and complementarity even under incomplete views. In addition, existing AP techniques are largely heuristic and lack theoretical grounding, making it difficult to

balance the removal of adversarial perturbations with the preservation of task-relevant signals. The proposed MAP framework redefines adversarial defense from the perspective of the Multi-view Information Bottleneck (MIB) (Federici et al., 2020) principle. We conceptualize adversarial attacks as the injection of a malicious, task-misaligned signal, thereby framing purification as a principled task of signal separation. In summary, our contributions are as follows:

**Theoretically**, we introduce Multi-view Adversarial Purification, the first information-theoretic paradigm for robust multi-view clustering. We formalize the defense as a principled signal separation problem, adapting the Multi-View Information Bottleneck principle to define a dual objective of maximizing informational sufficiency with the benign signal while enforcing purity against the adversarial noise.

**Algorithmically**, we propose GUARD (General Unsupervised Adversarial Robust Defense), a novel and practical framework that operationalizes the MAP paradigm. It features a unique, self-supervisory loss function that uses the downstream DMVC model as an oracle, creating an information bottleneck for adversarial noise as an emergent property of the optimization. The resulting purifier is inherently model-agnostic and task-aware.

**Experimentally**, we conduct extensive evaluations on five benchmark datasets, covering nine state-of-the-art complete and incomplete DMVC methods. The results demonstrate that GUARD not only significantly enhances adversarial robustness—often restoring performance to near-clean levels—but also offers a substantial efficiency advantage over powerful baselines like diffusion models.

Figure 1: GUARD: A universal, model-agnostic shield that intercepts adversarial examples, purifies their informational content to restore the benign data structure.

## 2 RELATED WORKS

### 2.1 DEEP MULTI-VIEW CLUSTERING

Numerous deep-learning-based approaches have been proposed for both complete and incomplete multi-view clustering. Among them, contrastive-learning-based methods (Trosten et al., 2023; Luo et al., 2024; Guo et al., 2024; Chao et al., 2024) and information-theoretic techniques (Huang et al., 2023; Yan et al., 2024; Xu et al., 2024; Zheng et al., 2025) have achieved strong performance by leveraging consistent and complementary information across views. A key inspiration for our work is the MIB principle (Federici et al., 2020), which provides a formal way to discard view-specific superfluous information to learn robust representations. However, these methods generally neglect adversarial robustness—a key requirement for reliable deployment in real-world applications. A handful of studies have explored trustworthy multi-view classification (Han et al., 2022; Lu et al., 2025; Liang et al., 2025), but they concentrate on uncertainty estimation across views rather than on adversarial robustness and cannot be directly applied to clustering.

To our knowledge, the most relevant to our work is only Huang et al. (2024) that has explicitly studied adversarially robust multi-view clustering. They first designed a multi-view attack to evaluate clustering robustness, then introduced the Adversarially Robust Deep Multi-View Clustering (AR-DMVC) framework, which combines *adversarial training* with a contrastive loss. While AR-DMVC represents an important step forward, its methodology introduces key limitations. As a model-specific adversarial training approach, it is inherently non-transferable, and its joint training objective often results in degraded performance on clean data. Furthermore, its application is restricted to complete multi-view scenarios. Our work addresses these gaps by proposing an input-level purification framework. Crucially, we are the first to provide a formal theoretical framework, grounded in MIB principles, for this purification task.

### 2.2 ADVERSARIAL ROBUST DEFENSE

**Adversarial training (AT)** is a widely used method to enhance the robustness of deep learning models by retraining them against adversarial attacks (Han et al., 2025). It involves augmenting the

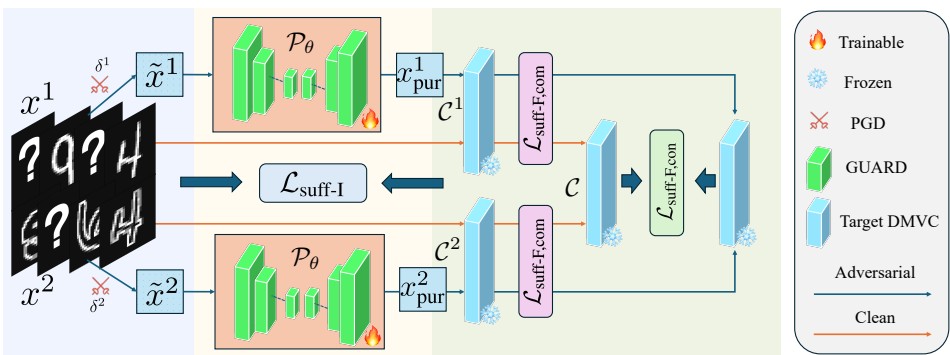

Figure 2: Illustration of GUARD's training pipeline.

training dataset with adversarial examples generated through various attack methods, such as FGSM (Goodfellow et al., 2014), PGD (Madry et al., 2018), and C&W (Carlini & Wagner, 2017). AT has been successfully applied to various domains, including but not limited to Adversarial Contrastive Learning (Xu et al., 2023b;c), Vision-Language Models (Schlarmann et al., 2024; Gong et al., 2025), and Multi-view Clustering (Huang et al., 2024).

**Adversarial Purification (AP)** is another common method that improves the robustness by removing adversarial perturbations from input data. Unlike AT, AP employs a pre-trained purification module without altering the target framework, exhibiting superior transferability and resilience to unseen attacks (Han et al., 2025). Notable AP techniques include self-supervised guided purification (Naseer et al., 2020; Shi et al., 2021; Bai et al., 2024), diffusion-based purification (Nie et al., 2022; Sun et al., 2025), and hybrid schemes combining AP with AT (Lin et al., 2024). While effective in single-view contexts, existing AP methods are ill-suited for multi-view clustering as they overlook the rich inter-view information structure. This highlights the need for a dedicated MAP approach. Our work is the first to offer such a principled solution by establishing an information-theoretic foundation for the MAP task.

## 3 METHODS

In this section, we introduce GUARD, a novel purification framework designed to operationalize the MAP paradigm introduced in Section 1. As illustrated in Figure 2, GUARD serves as a practical implementation of our theoretically-grounded MAP strategy. We first re-examine the purification task from an information-theoretic viewpoint, grounding our method in the principles of the MIB (Federici et al., 2020). We then demonstrate how the GUARD loss function serves as a tractable solution to this principled objective.

### 3.1 PROBLEM FORMULATION: AN INFORMATION-THEORETIC VIEW

Let $\{x^v\}_{v=1}^V$ denote the set of clean views and $\{\tilde{x}^v : \tilde{x}^v = x^v + \delta^v\}_{v=1}^V$ be their adversarially perturbed counterparts. Here, $\delta^v$ represents the view-specific, superfluous information (i.e., adversarial noise) that we aim to discard. The goal of a purification network $\mathcal{P}_\theta$ is to produce a purified output $x_{\text{pur}}^v = \mathcal{P}_\theta(\tilde{x}^v)$ that is both faithful to the original data and robust to the perturbation.

From an information bottleneck perspective, this translates to two primary goals:

**Goal 1. Sufficiency**: The purified output $x_{\text{pur}}^v$ must retain all task-relevant information contained in the original clean view $x^v$. This ensures that the utility of the data for the downstream clustering task is preserved. This can be expressed as maximizing the mutual information:

$$\max_\theta I(x_{\text{pur}}^v; x^v). \tag{1}$$

**Goal 2. Purity**: To achieve robustness, the purified output $x_{\text{pur}}^v$ must discard the adversarial information from the perturbation $\delta^v$. This creates a bottleneck for the noise. The objective is to minimize the conditional mutual information:

$$\min_\theta I(x_{\text{pur}}^v; \delta^v | x^v). \tag{2}$$

The key challenge arises because these two objectives are inherently in tension: the first pushes the representation to be maximally informative, while the second imposes a strict filter against adversarial perturbation. This tension motivates the central question of our method: *how to design a new loss function that maximizes one form of information while minimizing another?*

### 3.2 THE GUARD METHOD

To answer the central question of balancing these objectives, we design the GUARD loss function as a practical and effective proxy for the intractable mutual information terms. The GUARD loss, whose components are visually mapped out in Figure 2, is composed of three terms that collectively ensure the purified data $x_{\text{pur}}^v$ is a faithful representation of the clean data $x^v$ across multiple semantic levels.

**Reconstruction Sufficiency Term** ($\mathcal{L}_{\text{suff-I}}$): To ensure sufficiency at the most fundamental level, we must preserve the information present in the input signal itself. We introduce the Reconstruction Sufficiency Term, which measures the fidelity of the purified sample in the input (pixel) space. It is defined as:

$$\mathcal{L}_{\text{suff-I}} := \sum_{v=1}^{V} \left( \underbrace{||\mathcal{P}_{\theta^v}(\tilde{x}^v) - x^v||_2^2}_{\text{Corrective Fidelity}} + \underbrace{||\mathcal{P}_{\theta^v}(x^v) - x^v||_2^2}_{\text{Preservative Fidelity}} \right). \tag{3}$$

This term is composed of two critical components, each serving a distinct but complementary purpose: The first component, $||\mathcal{P}_{\theta^v}(\tilde{x}^v) - x^v||_2^2$, is the core purification objective. It compels the network to learn a mapping that actively removes the adversarial perturbation $\delta^v$ from the input $\tilde{x}^v$ and reconstructs the original clean signal $x^v$. This ensures the purifier is effective against attacks. The second component, $||\mathcal{P}_{\theta^v}(x^v) - x^v||_2^2$, acts as a regularizer for clean data. It ensures that the purifier does not unnecessarily distort unperturbed inputs, effectively forcing it to learn a near-identity mapping for benign samples. This is crucial for maintaining high performance on clean datasets and preventing the purifier from degrading the original data quality.

Together, these two components enforce a robust and consistent behavior: the purifier's output should always be a high-fidelity representation of the clean data, regardless of whether the input is adversarial or benign. This comprehensive fidelity guarantee serves as a powerful proxy for maximizing the mutual information $I(x_{\text{pur}}^v; x^v)$, ensuring that the foundational information of the original data is preserved under all conditions.

**Feature-Space Sufficiency Term** ($\mathcal{L}_{\text{suff-F}}$): While input-space fidelity is crucial, for downstream tasks like clustering, preserving the semantic information captured by deep features is paramount. The Feature-Space Sufficiency Term achieves this by ensuring the representations of the purified data align with those of the clean data in the latent space of the target DMVC model. This term combines two objectives: preserving view-specific (complementary) and shared (consistent) features.

First, the Complementary Feature Sufficiency loss ensures that unique characteristics of each view are retained:

$$\mathcal{L}_{\text{suff-F,com}} := \sum_{v=1}^{V} \mathcal{D}(\mathcal{C}^v(x^v), \mathcal{C}^v(x_{\text{pur}}^v)), \tag{4}$$

where $x_{\text{pur}}^v = \mathcal{P}_{\theta^v}(\tilde{x}^v)$, $\mathcal{C}^v$ is the target DMVC view-specific encoder for view $v$, and $\mathcal{D}(\cdot, \cdot)$ denotes a feature-distance metric (e.g., Euclidean, Wasserstein, or Cosine distance). Second, the Consistent Feature Sufficiency loss ensures the shared information across all views is preserved:

$$\mathcal{L}_{\text{suff-F,con}} := \mathcal{D}(\mathcal{C}(\{x^v\}_{v=1}^{V}), \mathcal{C}(\{x_{\text{pur}}^v\}_{v=1}^{V})). \tag{5}$$

where $\mathcal{C}$ is the target DMVC fusion encoder that captures the shared information across all views, and $\{x^v\}_{v=1}^{V}$ denotes the set of all views. Minimizing the distance between latent representations, as done in Eqs. 4 and 5, is a cornerstone of contrastive learning and self-supervised methods, which are known to be effective because they maximize the mutual information between different views or augmentations of the data (Federici et al., 2020). Therefore, our feature-space terms act as a proxy for maximizing the mutual information in the semantic feature space, $I(\mathcal{C}(x_{\text{pur}}); \mathcal{C}(x))$. Thus, the entire explicit objective of GUARD, $\mathcal{L}_{\mathcal{P}_\theta}$, can be seen as a multi-faceted strategy to solve the sufficiency problem:

$$\arg\min_{\theta} \mathcal{L}_{\mathcal{P}_\theta} \implies \arg\max_{\theta} I(x_{\text{pur}}; x^v). \tag{6}$$

**Purity as an Emergent Property.** The bottleneck is not an explicit term in our loss function but rather an emergent property of the learning dynamics. The purifier network $\mathcal{P}_\theta$ takes the perturbed data $\tilde{x}^v = x^v + \delta^v$ as input. However, the entire supervisory signal provided by our loss function, $\mathcal{L}_{\mathcal{P}_\theta}$, is derived exclusively from the clean data $x^v$. There is no component in the loss that rewards the network for preserving any information about the perturbation $\delta^v$.

Consequently, during optimization, the network $\mathcal{P}_\theta$ is incentivized to become a function that is maximally sensitive to the aspects of its input that correlate with $x^v$, and minimally sensitive to the parts that do not—namely, the perturbation $\delta^v$. For a network with finite capacity, the most efficient way to maximize the reconstruction of $x^v$ is to learn to be invariant to $\delta^v$, effectively discarding it.

Therefore, the optimization process implicitly solves the minimality objective:

$$\arg\min_\theta \mathcal{L}_{\mathcal{P}_\theta} \implies \arg\min_\theta I(\mathcal{P}_\theta(x^v + \delta^v); \delta^v | x^v). \tag{7}$$

In summary, GUARD establishes an information bottleneck not by adding a penalty term, but by structuring the optimization landscape such that the only path to success for the purifier is to learn to discard the superfluous adversarial information.

**Overall Objective.** The final GUARD loss is a weighted sum of the sufficiency objectives in both the input and feature spaces. By pursuing sufficiency so comprehensively, the loss function creates an optimization landscape where the purifier must implicitly learn to discard the non-supervisory adversarial noise to succeed. The total objective is:

$$\mathcal{L}_{\mathcal{P}_\theta} = \alpha \mathcal{L}_{\text{suff-F,com}} + \beta \mathcal{L}_{\text{suff-F,con}} + \gamma \mathcal{L}_{\text{suff-I}}, \tag{8}$$

where $\alpha, \beta, \gamma$ are hyperparameters that balance the preservation of information across the different semantic levels. In conclusion, all explicit terms in the GUARD loss contribute to the **sufficiency** objective by maximizing the mutual information between the purified output ($x_{\text{pur}}$) and the clean data ($x$) in different spaces. The **purity** objective, or the information bottleneck, emerges implicitly from this optimization process. Furthermore, we provide a formal derivation in Appendix A.2, which shows that minimizing our sufficiency loss terms is equivalent to maximizing a variational lower bound on mutual information.

To implement the purification module $\mathcal{P}_\theta$, we adopt the UNet architecture (Ronneberger et al., 2015), whose symmetric encoder–decoder structure with skip connections enables effective aggregation of multi-scale features and faithful reconstruction of spatial details. This design allows the network to leverage both global context and fine-grained information when removing localized adversarial perturbations, and its proven success in image restoration and denoising tasks makes it particularly well suited for learning precise (Ho et al., 2020; Rombach et al., 2022). In the training phase, the adversarial example $\tilde{x}_i^v$ is generated by finding a worst-case perturbation within an $\ell_\infty$-ball of radius $\epsilon$ that maximally degrades this informational sufficiency:

$$\tilde{x}^v \in \arg\max_{x' \in \mathcal{B}_\epsilon[x^v]} \mathcal{L}_{\text{target}}(x'; \vartheta), \tag{9}$$

where $\mathcal{B}_\epsilon[x^v] = \{x' : \|x' - x^v\|_\infty \leq \epsilon\}$, and $\mathcal{L}_{\text{target}}(\cdot; \vartheta)$ is the (unsupervised) DMVC loss parameterized by $\vartheta$. We use the Projected Gradient Descent (PGD) method (Madry et al., 2018) to solve this optimization problem. The learning algorithm of GUARD has been provided in Appendix A.3.

## 4 EXPERIMENTS

Following (Trosten et al., 2023; Huang et al., 2024), we evaluate the performance of our proposed GUARD on five multi-view datasets: EdgeMNIST, EdgeFashion, NoisyMNIST, NoisyFashion, and RegDB. The first four datasets are generated from MNIST and FashionMNIST, which are widely used in the multi-view clustering community. The last dataset is RegDB, which is a real-world dataset for multi-view clustering. For the complete DMVC methods, we target six methods: EAMC (Zhou & Shen, 2020), SiMVC (Trosten et al., 2021), CoMVC (Trosten et al., 2021), InfoDDC (Trosten et al., 2023), AR-DMVC and AR-DMVC-AM (Huang et al., 2024). About the incomplete DMVC methods, we chose the APADC (Xu et al., 2023a), DVIMC (Xu et al., 2024) and LOGIC (Zheng

et al., 2025). All the target DMVC methods are open-sourced and implemented in PyTorch. We use the same backbone networks as the original papers, and all models are trained with the same hyperparameters. Due to the limited space, more details about the target methods, datasets and training process can be found in the Appendix A.4,A.5,A.6. We mainly employ the adversarial DMVC attack framework introduced in Huang et al. (2024), which—to the best of our knowledge—is the sole method specifically designed to generate multi-view adversarial perturbations. In the experiments, we set the $\epsilon$ equal to the attack threshold and report clustering accuracy (**ACC**), normalized mutual information (**NMI**), and **Purity** as evaluation metrics.

## 4.1 Adversarial Defense Performance on Deep Multi-view Clustering

Table 1 presents the pre-attack (PRE), post-attack (POST), and post-purification (Ours) performance of various DMVC methods across five benchmark datasets. We set the adversarial attack threshold to 0.3 for EdgeMNIST, EdgeFashion, and NoisyMNIST, 0.2 for RegDB, and 0.15 for NoisyFashion, following the protocol in (Huang et al., 2024). For incomplete DMVC experiments, we set the data existence rate to 0.7 (i.e., 30% missing data); additional results with existence rates of 0.5 and 0.3 are provided in the Appendix A.7. We have the following observations:

1) The proposed PUR step restores complete methods to near their PRE levels—e.g., AR-DMVC on NoisyMNIST rebounds from POST 0.61 to PUR 1.00, and SiMVC recovers from 0.22→0.71 ACC on EdgeMNIST. By comparison, incomplete approaches show more modest gains: APADC on EdgeMNIST recovers only from 0.11→0.40 ACC, DVIMC on RegDB from 0.39→0.48, and LOGIC typically achieves 60–75% recovery. This disparity underscores the restorative ceiling imposed by missing views.

2) Although models incorporating adversarial examples during training (AR-DMVC series) exhibit the strongest resilience under attack, appending our post-purification module not only further elevates their robustness but also empowers conventional DMVC models to exceed AR-DMVC performance. For instance, on EdgeMNIST, EAMC enhanced with our purifier outperforms the AR-DMVC baseline. This demonstrates that our purification strategy not only augments state-of-the-art adversarial defenses but is also readily transferable and extensible to a wide spectrum of off-the-shelf DMVC frameworks.

3) In most cases, integrating AR-DMVC variants with our PUR step can achieve the best performance (e.g., EdgeFashion, NoisyMNIST, NoisyFashion), thereby providing a balanced and highly effective framework for enhancing robustness in deep multi-view clustering.

The visualizations of the purified images are supplemented in the Appendix A.16. They illustrate the effectiveness of GUARD in restoring adversarially perturbed images to a state closer to the original images, thereby enhancing the robustness of DMVC models against adversarial attacks. In addition, we further validate GUARD's resilience against the classic white-box FGSM attack in Appendix A.14. This demonstrates GUARD's versatility and effectiveness in defending against various types of adversarial attacks.

## 4.2 Comparison with Diffusion-based Purification Baseline

To benchmark our framework against existing purification paradigms, we compare it with a strong and widely-used baseline: a **pre-trained diffusion model**. As dedicated multi-view purifiers are not readily available, we implemented a powerful single-view baseline inspired by Nie et al. (2022). Specifically, we used a publicly available diffusion model pre-trained on MNIST and applied it as a purifier to each view of the EdgeMNIST dataset independently. This provides a robust, off-the-shelf baseline for the purification task. The comparative results for all nine DMVC methods are presented in Figure 3a. The performance of the diffusion purifier is denoted as PUR (Diffpure), and our method is PUR (Ours).

Our analysis of these results is twofold. First, regarding clustering performance, the pre-trained diffusion model proves to be a strong baseline, achieving comparable or superior results for certain methods (e.g., CoMVC, DVIMC). However, our proposed framework, which is trained from scratch on the task data, outperforms the diffusion baseline on a clear majority (6 out of 9) of the DMVC methods. The performance gap is particularly notable on recent and complex architectures like InfoDDC and AR-DMVC, highlighting the benefits of our task-specific, multi-view-aware design.

Table 1: Pre-attack (PRE), post-attack (POST) and post-purification (**Ours**) performance for DMVC methods on five datasets. EAMC to AR-DMVC-AM are complete DMVC methods, while APADC, DVIMC and LOGIC are incomplete DMVC methods.

| Method | Phase | EdgeMNIST | | | EdgeFashion | | | NoisyMNIST | | | NoisyFashion | | | RegDB | | |
|---|---|---|---|---|---|---|---|---|---|---|---|---|---|---|---|---|
| | | ACC | NMI | Purity | ACC | NMI | Purity | ACC | NMI | Purity | ACC | NMI | Purity | ACC | NMI | Purity |
| EAMC (CVPR'20) | PRE | 0.75 | 0.78 | 0.79 | 0.50 | 0.53 | 0.53 | 0.80 | 0.90 | 0.83 | 0.49 | 0.62 | 0.50 | 0.51 | 0.62 | 0.56 |
| | POST | *0.51* | *0.47* | *0.54* | *0.17* | *0.23* | *0.20* | *0.18* | *0.15* | *0.21* | *0.21* | *0.21* | *0.25* | *0.34* | *0.39* | *0.40* |
| | **Ours** | 0.75 | 0.78 | 0.79 | 0.34 | 0.31 | 0.36 | 0.66 | 0.67 | 0.67 | 0.48 | 0.60 | 0.49 | 0.62 | 0.68 | 0.66 |
| SiMVC (CVPR'21) | PRE | 0.71 | 0.74 | 0.75 | 0.59 | 0.53 | 0.60 | 0.87 | 0.94 | 0.91 | 0.52 | 0.54 | 0.55 | 0.87 | 0.90 | 0.87 |
| | POST | *0.22* | *0.20* | *0.24* | *0.33* | *0.29* | *0.34* | *0.17* | *0.21* | *0.23* | *0.22* | *0.23* | *0.23* | *0.76* | *0.84* | *0.77* |
| | **Ours** | 0.71 | 0.71 | 0.73 | 0.45 | 0.42 | 0.47 | 0.85 | 0.89 | 0.88 | 0.33 | 0.33 | 0.35 | 0.86 | 0.90 | 0.87 |
| CoMVC (CVPR'21) | PRE | 0.65 | 0.68 | 0.68 | 0.51 | 0.48 | 0.53 | 0.88 | 0.95 | 0.91 | 0.66 | 0.69 | 0.70 | 0.60 | 0.68 | 0.65 |
| | POST | *0.42* | *0.39* | *0.44* | *0.10* | *0.00* | *0.10* | *0.11* | *0.00* | *0.11* | *0.29* | *0.31* | *0.30* | *0.43* | *0.44* | *0.46* |
| | **Ours** | 0.64 | 0.68 | 0.67 | 0.38 | 0.33 | 0.39 | 0.78 | 0.76 | 0.78 | 0.62 | 0.64 | 0.67 | 0.58 | 0.67 | 0.64 |
| InfoDDC (CVPR'23) | PRE | 0.50 | 0.53 | 0.56 | 0.45 | 0.40 | 0.48 | 0.92 | 0.94 | 0.92 | 0.43 | 0.37 | 0.43 | 0.50 | 0.60 | 0.54 |
| | POST | *0.37* | *0.35* | *0.42* | *0.12* | *0.04* | *0.12* | *0.11* | *0.00* | *0.11* | *0.21* | *0.16* | *0.22* | *0.48* | *0.69* | *0.58* |
| | **Ours** | 0.49 | 0.50 | 0.55 | 0.26 | 0.21 | 0.26 | 0.68 | 0.61 | 0.68 | 0.25 | 0.19 | 0.26 | 0.56 | 0.67 | 0.60 |
| AR-DMVC (ICML'24) | PRE | 0.62 | 0.68 | 0.64 | 0.57 | 0.53 | 0.59 | 1.00 | 1.00 | 1.00 | 0.57 | 0.58 | 0.63 | 0.56 | 0.68 | 0.61 |
| | POST | *0.47* | *0.49* | *0.48* | *0.40* | *0.36* | *0.41* | *0.61* | *0.64* | *0.61* | *0.54* | *0.54* | *0.55* | *0.47* | *0.52* | *0.52* |
| | **Ours** | 0.62 | 0.67 | 0.63 | 0.44 | 0.40 | 0.45 | 1.00 | 0.98 | 1.00 | 0.57 | 0.57 | 0.62 | 0.55 | 0.63 | 0.59 |
| AR-DMVC-AM (ICML'24) | PRE | 0.67 | 0.73 | 0.72 | 0.61 | 0.57 | 0.62 | 0.84 | 0.93 | 0.88 | 0.71 | 0.69 | 0.71 | 0.64 | 0.73 | 0.69 |
| | POST | *0.64* | *0.69* | *0.70* | *0.29* | *0.27* | *0.30* | *0.66* | *0.73* | *0.69* | *0.34* | *0.48* | *0.37* | *0.49* | *0.55* | *0.52* |
| | **Ours** | 0.66 | 0.72 | 0.71 | 0.45 | 0.43 | 0.46 | 0.85 | 0.91 | 0.88 | 0.65 | 0.62 | 0.66 | 0.64 | 0.70 | 0.68 |
| APADC (TIP'23) | PRE | 0.42 | 0.36 | 0.45 | 0.32 | 0.27 | 0.33 | 0.92 | 0.85 | 0.92 | 0.70 | 0.70 | 0.72 | 0.46 | 0.34 | 0.46 |
| | POST | *0.11* | *0.01* | *0.12* | *0.10* | *0.00* | *0.10* | *0.13* | *0.04* | *0.14* | *0.10* | *0.00* | *0.10* | *0.37* | *0.36* | *0.42* |
| | **Ours** | 0.40 | 0.26 | 0.41 | 0.21 | 0.15 | 0.24 | 0.66 | 0.51 | 0.66 | 0.55 | 0.48 | 0.58 | 0.45 | 0.37 | 0.48 |
| DVIMC (AAAI'24) | PRE | 0.56 | 0.58 | 0.62 | 0.56 | 0.56 | 0.60 | 0.84 | 0.88 | 0.86 | 0.64 | 0.64 | 0.67 | 0.76 | 0.78 | 0.79 |
| | POST | *0.34* | *0.24* | *0.36* | *0.31* | *0.26* | *0.36* | *0.27* | *0.16* | *0.28* | *0.34* | *0.29* | *0.36* | *0.39* | *0.51* | *0.46* |
| | **Ours** | 0.51 | 0.43 | 0.51 | 0.51 | 0.52 | 0.55 | 0.54 | 0.49 | 0.57 | 0.49 | 0.45 | 0.50 | 0.48 | 0.54 | 0.51 |
| LOGIC (NN'25) | PRE | 0.50 | 0.43 | 0.52 | 0.47 | 0.45 | 0.50 | 0.52 | 0.42 | 0.53 | 0.45 | 0.34 | 0.46 | 0.49 | 0.57 | 0.57 |
| | POST | *0.31* | *0.19* | *0.33* | *0.35* | *0.26* | *0.36* | *0.42* | *0.33* | *0.42* | *0.36* | *0.31* | *0.39* | *0.37* | *0.49* | *0.39* |
| | **Ours** | 0.47 | 0.37 | 0.48 | 0.45 | 0.34 | 0.46 | 0.51 | 0.41 | 0.53 | 0.46 | 0.36 | 0.46 | 0.47 | 0.54 | 0.48 |

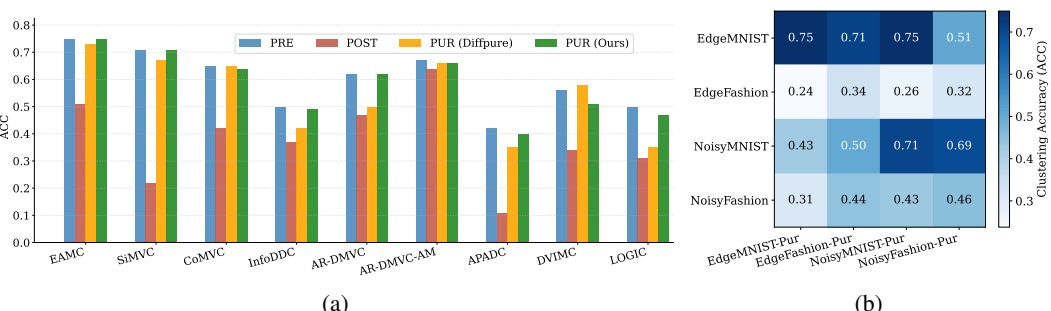

(a)                                    (b)

Figure 3: (a) Comparison with a pre-trained Diffusion model purifier on the EdgeMNIST dataset ($\epsilon = 0.3$). (b) Transfer matrix of EAMC for purifiers: each column denotes the source dataset used to train the purifier, and each row denotes the target dataset on which ACC is evaluated.

Second, we analyze the computational cost, a critical factor for practical applications. The diffusion purifier relies on a time-consuming iterative denoising process, resulting in a substantial difference in inference time. Specifically, the diffusion purifier required an average of ˜**568.58 seconds** per batch, whereas our method took only ˜**1.46 seconds**. This nearly **400-fold speedup** makes our framework a substantially more practical and scalable solution. In conclusion, this comparative analysis demonstrates that our framework offers a superior balance of high performance across a wide range of modern DMVC models and vastly greater computational efficiency.

### 4.3 Cross Datasets & Models Purification

As shown in the Figure 3b, although each purifier achieves its peak performance when applied to the very dataset on which it was trained, we also observe meaningful gains when a purifier is used on a different dataset. For instance, the NoisyMNIST-trained purifier still delivers strong clustering on EdgeMNIST (ACC=0.75). Such off-diagonal successes demonstrate that our purification model has indeed learned invariant representations that transfer at least partially across domains. Taken

together, these results confirm not only the importance of in-domain training but also the preliminary cross-dataset transferability of our adversarial purification network.

In Figure 4, we demonstrate the cross-model transferability of our GUARD framework across multiple complete and incomplete DMVC methods. The results reveal that most purifiers achieve their highest accuracy on their native source model (diagonal entries) while still maintaining competitively high ACC when deployed on heterogeneous target models—most notably the LOGIC purifiers. This robust performance indicates that the learned purification transformations capture the intrinsic, model-agnostic characteristics of adversarial perturbations.

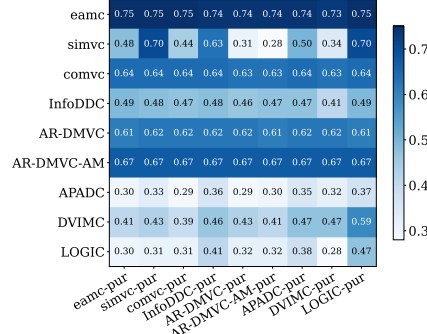

### 4.4 ABLATION
STUDY OF SUFFICIENCY COMPONENTS

Figure 5a illustrates the ACC attained by activating each GUARD component in isolation versus the full model across nine target methods. In this experiment, we instantiate the full GUARD framework by setting the weighting hyperparameters $\alpha$, $\beta$ and $\gamma$ all to 1; each ablation variant

Figure 4: Transfer matrix of ACC for purifiers: each column denotes the source model used to train the purifier, and each row denotes the target model on which ACC is evaluated.

then retains exactly one of these components (weight = 1) while zeroing out the other two. The grid search results for these hyperparameters are provided in the Appendix A.11.

The results underscore the importance of our multi-faceted approach to ensuring informational Sufficiency. We observe that enforcing Sufficiency only in the input space ($\gamma = 1$) provides a reasonable performance baseline, but is insufficient on its own for more complex methods. Conversely, enforcing Sufficiency in the consistent feature space ($\beta = 1$) is critical for preserving the semantic structure required by the clustering models. The superior performance of the full GUARD model (blue bar) across nearly all methods demonstrates that a comprehensive Sufficiency objective, which ensures fidelity at both the pixel and semantic levels, creates the most effective learning environment. This comprehensive objective provides the strongest possible guidance for the purifier, which in turn enables the emergent information bottleneck to effectively discard the adversarial perturbations.

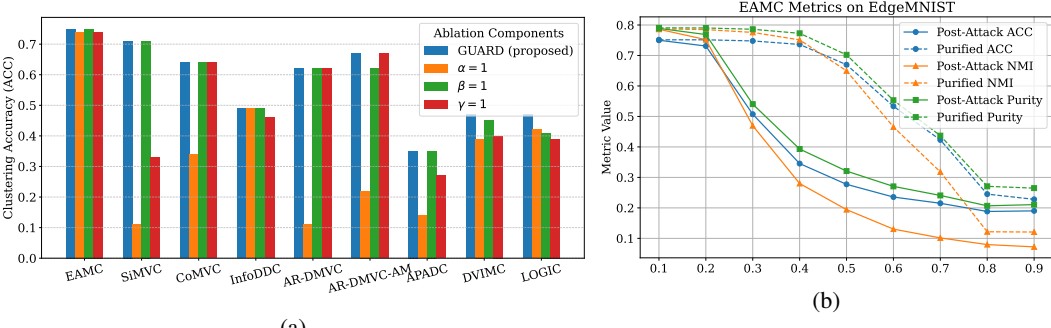

(a)

(b)

Figure 5: (a) Ablation study of the influence of the three components in our proposed GUARD framework. The results are reported on EdgeMNIST with ACC. (b) EAMC vs. attack strengths on EdgeMNIST.

### 4.5 INFLUENCE OF ATTACK STRENGTHS, DIFFERENT $\mathcal{D}$, AND EXISTING RATE

Figure 5b illustrates how ACC, NMI, and Purity evolve for both EAMC and SiMVC on EdgeMNIST as the attack perturbation increases (from 0.1 to 0.9), juxtaposing raw post-attack performance with results obtained after purification. We can observe that as attack strength grows, the purified curves for both models consistently exceed their post-attack counterparts, demonstrating the purifier's ability to mitigate adversarial degradation. Furthermore, the gap between purified and unpurified performance widens from 0.2 to 0.4 attack strengths, indicating that our purification pipeline scales effectively with increasing attack severity. These results empirically validate the efficacy of the GUARD in

preserving clustering performance under escalating adversarial stress. More results of other methods can be found in the Appendix A.10.

Table 2 illustrates the post-purification ACC for Cosine, Euclidean, and Wasserstein metrics across all evaluated methods. The NMI and Purity metrics results can be found in the Appendix A.13. Euclidean distance achieves marginally higher peak ACC in most cases but exhibits greater variability in APADC. Wasserstein distance generally underperforms, with several methods experiencing notable drops in accuracy relative to Cosine. Overall, **Cosine** distance provides the most consistent and robust purification outcomes, justifying its selection as the default $\mathcal{D}$ metric in Eqs. (4) and (5).

Table 2: Post-purification ACC of DMVC methods on EdgeMNIST with $\epsilon = 0.3$, comparing three different distance metrics.

| Method | Cosine | Euclidean | Wasserstein |
|---|---|---|---|
| EAMC | **0.74** | **0.74** (+0.00) | **0.74** (+0.00) |
| SiMVC | 0.70 | **0.71** (+0.01) | 0.33 (-0.37) |
| CoMVC | 0.64 | **0.65** (+0.01) | 0.64 (+0.00) |
| InfoDDC | 0.48 | **0.49** (+0.01) | 0.44 (-0.04) |
| AR-DMVC | **0.62** | **0.62** (+0.00) | 0.50 (-0.12) |
| AR-DMVC-AM | **0.67** | 0.64 (-0.03) | **0.67** (+0.00) |
| APADC | **0.40** | 0.33 (-0.07) | 0.37 (-0.03) |
| DVIMC | 0.47 | **0.48** (+0.01) | 0.38 (-0.09) |
| LOGIC | **0.47** | 0.45 (-0.02) | 0.39 (-0.08) |

Table 3: Performance of incomplete methods under varying data existing rates on EdgeMNIST.

| Method | Phase | Rate 0.7 | | | Rate 0.5 | | | Rate 0.3 | | |
|---|---|---|---|---|---|---|---|---|---|---|
| | | ACC | NMI | Purity | ACC | NMI | Purity | ACC | NMI | Purity |
| APADC | PRE | 0.42 | 0.36 | 0.45 | 0.37 | 0.27 | 0.37 | 0.36 | 0.28 | 0.39 |
| | POST | 0.11 | 0.01 | 0.12 | 0.11 | 0.00 | 0.11 | 0.11 | 0.00 | 0.11 |
| | **Ours** | 0.40 | 0.26 | 0.41 | 0.36 | 0.24 | 0.37 | 0.33 | 0.28 | 0.38 |
| DVIMC | PRE | 0.60 | 0.58 | 0.65 | 0.58 | 0.58 | 0.62 | 0.58 | 0.57 | 0.62 |
| | POST | 0.26 | 0.16 | 0.28 | 0.24 | 0.16 | 0.26 | 0.23 | 0.13 | 0.24 |
| | **Ours** | 0.49 | 0.40 | 0.52 | 0.46 | 0.39 | 0.48 | 0.44 | 0.36 | 0.46 |
| LOGIC | PRE | 0.50 | 0.43 | 0.52 | 0.49 | 0.42 | 0.51 | 0.54 | 0.44 | 0.54 |
| | POST | 0.33 | 0.22 | 0.34 | 0.23 | 0.13 | 0.24 | 0.17 | 0.08 | 0.18 |
| | **Ours** | 0.47 | 0.37 | 0.48 | 0.44 | 0.34 | 0.44 | 0.35 | 0.29 | 0.40 |

Table 3 reports the clustering results of three incomplete DMVC methods (APADC, DVIMC and LOGIC) at existing rate of 0.7, 0.5 and 0.3. Most methods exhibit a clear decline as the existing rate decreases, underscoring the adverse effect of missing views on clustering performance. In addition, our proposed purifier consistently boosts performance across all existing rates, effectively narrowing the gap induced by higher missing ratios. In the Appendix A.12, we further explore the performance of our purifier under extreme missing rates. These findings confirm that the proposed purifier is useful and robust under varying degrees of data incompleteness.

## 5 CONCLUSION

This paper introduces Multi-view Adversarial Purification, a novel information-theoretic defense paradigm to address the critical problem of adversarial vulnerability in DMVC. We conceptualize adversarial purification as a principled signal separation task, guided by the Multi-View Information Bottleneck, with a dual objective: maximizing informational sufficiency with benign data while enforcing purity against adversarial noise. We introduce GUARD, a model-agnostic multi-view purification framework. Crucially, GUARD's unique self-supervisory design enables the information bottleneck to emerge naturally during optimization, effectively filtering adversarial perturbations without explicit constraints. This allows GUARD to serve as a seamlessly integrable front-end defense for any DMVC model. Extensive experiments on diverse datasets and DMVC methods demonstrate GUARD's superior performance. It consistently restores clustering accuracy to near-clean levels and outperforms diffusion-based purification baselines, achieving up to a 400x speedup in inference. Our analyses across varying attack strengths, data completion rates, and threat models further underscore the robustness and versatility of our proposed method.

ETHICS STATEMENT

This study relies on algorithmic methods and available public data, without human participation or sensitive information usage. Operating under ICLR ethical frameworks with no institutional conflicts, we promote judicious application of results and maintain methodological openness for scientific reproducibility.

REPRODUCIBILITY STATEMENT

An anonymous link of the source code is provided in Appendix A.

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

# A APPENDIX

The source code for GUARD is provided in an anonymous repository[1]. Developed within the PyTorch framework, the code is engineered for straightforward adaptation to various deep multi-view clustering methods. The repository contains detailed instructions for installation and usage, complemented by examples illustrating the application of GUARD to diverse datasets and models.

---

[1] https://anonymous.4open.science/r/GUARD-7535

A.1 STATEMENT OF THE USE OF LARGE LANGUAGE MODELS (LLMS)

In this paper, we just used the LLM, ChatGPT, to polish the language of the paper. We did not use LLMs to generate any content or ideas in this work. We have verified the accuracy of all content and ideas in the paper.

## A.2 THEORETICAL JUSTIFICATION OF THE SUFFICIENCY PROXIES

In this section, we provide a formal justification for our sufficiency-driven loss design, linking the minimization of reconstruction and feature-space distances to the information-theoretic objective of maximizing mutual information.

**Theorem A.1.** *The sufficiency objective of GUARD, which seeks to maximize the mutual information $I(x_{pur}^v; x^v)$ between the purified output $x_{pur}^v$ and the clean data $x^v$, is achieved by minimizing the sufficiency loss terms $\mathcal{L}_{suff-I}$ and $\mathcal{L}_{suff-F}$. Minimizing these terms is equivalent to maximizing a variational lower bound on the mutual information.*

*Proof.* Our proof is structured in two parts, addressing the two critical levels of the information pipeline where sufficiency is enforced: the input space and the semantic feature space.

**Part 1: Reconstruction Sufficiency.** The relationship between reconstruction error and mutual information is well-established in rate-distortion theory. Minimizing the Mean Squared Error (MSE) between a reconstruction $x_{\text{pur}}^v$ and an original signal $x^v$ is equivalent to maximizing a lower bound on their mutual information, especially under a Gaussian channel assumption.

Our Reconstruction Sufficiency Term, $\mathcal{L}_{\text{suff-I}}$, leverages this principle by jointly minimizing two distinct MSE objectives, as defined in Eq. 3:

- The **Corrective Fidelity** term, $||\mathcal{P}_{\theta^v}(\tilde{x}^v) - x^v||_2^2$, minimizes the MSE between the purified adversarial image and the original clean image. This directly trains the purifier to remove the adversarial perturbation and recover the benign signal.

- The **Preservative Fidelity** term, $||\mathcal{P}_{\theta^v}(x^v) - x^v||_2^2$, minimizes the MSE when the input is already clean. This component acts as a crucial regularizer, ensuring that the purifier learns a near-identity function for non-adversarial data and does not introduce distortions.

By minimizing the sum of these two reconstruction errors, our framework ensures that the output $x_{\text{pur}}^v$ is a high-fidelity representation of the clean signal $x^v$ *regardless of the input's condition*. This comprehensive fidelity guarantee serves as a robust proxy for maximizing the mutual information $I(x_{\text{pur}}^v; x^v)$.

**Part 2: Feature-Space Sufficiency.** By the Data Processing Inequality, we have $I(x_{\text{pur}}^v; x^v) \geq I(\mathcal{C}(x_{\text{pur}}); \mathcal{C}(x))$, meaning that maximizing the mutual information between features contributes to the overall sufficiency objective. We demonstrate this principle using the Complementary Feature Sufficiency term as a representative example.

Let $z^v = \mathcal{C}^v(x^v)$ and $z_{\text{pur}}^v = \mathcal{C}^v(x_{\text{pur}}^v)$ be the view-specific features for view $v$. The goal is to maximize $I(z_{\text{pur}}^v; z^v)$, which is equivalent to minimizing the conditional entropy $H(z^v|z_{\text{pur}}^v)$. We can establish a tractable upper bound on this entropy by introducing a variational approximation $q_\phi(z|z_{\text{pur}})$:

$$H(z^v|z_{\text{pur}}^v) \leq \mathbb{E}_{p(z^v, z_{\text{pur}}^v)}[-\log q_\phi(z^v|z_{\text{pur}}^v)]. \tag{10}$$

By parameterizing this distribution as an isotropic Gaussian, $q_\phi(z^v|z_{\text{pur}}^v) = \mathcal{N}(z^v|z_{\text{pur}}^v, \sigma^2 I)$, minimizing the negative log-likelihood becomes equivalent to minimizing the expected squared Euclidean distance:

$$\min_\theta \mathbb{E}\left[||z^v - z_{\text{pur}}^v||_2^2\right]. \tag{11}$$

This formally shows that minimizing $\mathcal{L}_{\text{suff-F,com}}$ maximizes a lower bound on the mutual information for complementary features. The exact same principle applies to the Consistent Feature Sufficiency

---

**Algorithm 1** Training Procedure for the GUARD

---

1: **Input:** Unlabeled training set $U$, traget DMVC model $\mathcal{C}$, total training epochs $E$, batch size $B$, adversarial budget $\epsilon > 0$.
2: **Output:** The purification model $\mathcal{P}$.
3: Initialize $\theta$ for $\mathcal{P}_\theta$
4: **for** $e = 0$ **to** $E - 1$ **do**
5:    **for** batch $m = 1, \ldots, \lceil |U|/B \rceil$ **do**
6:       Sample a minibatch $B_m$ from $U$.
7:       Generate adversarial samples via Eq. 9.
8:       Compute the GUARD loss $\mathcal{L}_{\mathcal{P}_\theta}$ via Eq. 8.
9:       Update parameters: $\theta \leftarrow \theta - \eta \nabla_\theta \mathcal{L}_{\mathcal{P}_\theta}$.
10:    **end for**
11: **end for**

---

term, $\mathcal{L}_{\text{suff-F,con}}$, where minimizing the distance between fused feature representations maximizes their corresponding mutual information.

By jointly minimizing both $\mathcal{L}_{\text{suff-I}}$ and $\mathcal{L}_{\text{suff-F}}$, the overall GUARD sufficiency objective effectively maximizes a lower bound on the mutual information $I(x_{\text{pur}}^v; x^v)$ at both the input and feature levels, thus proving the theorem. While the derivation for the feature space is shown for Euclidean distance, the underlying principle extends to other metrics like Cosine distance that enforce feature similarity.

$\square$

### A.3   ALGORITHM

We supplement the training procedure of our proposed GUARD in Algorithm 1.

### A.4   DATASET STATISTICS

In this section, we provide the statistics of the datasets used in our experiments. The datasets include EdgeMNIST, EdgeFashion, NoisyMNIST, NoisyFashion, and RegDB. The number of samples, dimensions, and clusters for each dataset is listed in Table 4 below.

Table 4: Description of datasets.

| Dataset | Samples | Dimensions | Clusters |
|---|---|---|---|
| EdgeMNIST | 70000 | $28 \times 28$ | 10 |
| EdgeFashion | 70000 | $28 \times 28$ | 10 |
| NoisyMNIST | 70000 | $28 \times 28$ | 10 |
| NoisyFashion | 70000 | $28 \times 28$ | 10 |
| RegDB | 1000 | $3 \times 128 \times 64$ | 10 |

### A.5   SOURCE CODE AND ANALYSIS OF TARGET METHODS

The source code of complete DMVC methods are provided by [2], the AR-DMVC and AR-DMVC-AM are from [3], and the incomplete DMVC methods, APADC [4], DVIMC [5], and LOGIC [6] are also available in their respective repositories.

Compared to the pioneering AR-DMVC framework for adversarial multi-view clustering, our proposed GUARD method offers several *key advantages*:

---

[2] https://github.com/DanielTrosten/DeepMVC/tree/main/src/models
[3] https://github.com/libertyhhn/AR-DMVC/tree/main/models
[4] https://github.com/SubmissionsIn/APADC
[5] https://github.com/tkkxgh/DVIMC-pytorch
[6] https://github.com/YanghangZheng-GDUT/LOGIC

- **Input-Level Purification Preserving Clean-Data Performance.** GUARD applies adversarial defense directly to each view's input without altering the clustering architecture or training procedure, thereby fully preserving accuracy on unperturbed data. By contrast, AR-DMVC's joint adversarial training and architectural modifications incur a measurable drop in clustering performance on clean datasets.

- **Model-Agnostic Transferability.** Our purifier can be seamlessly applied to any off-the-shelf multi-view clustering algorithm. In contrast, AR-DMVC is tightly coupled to a particular complete-view clustering scheme, making it cumbersome to adapt to alternative architectures or objectives.

- **Support for Incomplete DMVC.** GUARD's unsupervised purification loss naturally handles arbitrary missing-view patterns by operating only on available views. By contrast, AR-DMVC assumes all views are present at inference time and fails when faced with incomplete data.

## A.6 DETAILS OF THE TRAINING AND THE PURIFICATION NETWORK ARCHITECTURE

All datasets are first shuffled using a fixed random seed to ensure reproducibility, then split evenly into training and test sets. For example, EdgeMNIST comprises 70 000 examples—35 000 are used to train the purification and clustering models, and the remaining 35 000 are held out for adversarial attack experiments and subsequent purification. Importantly, our entire pipeline is unsupervised: no label information is ever used during training. We train with a batch size of 2048 on EdgeMNIST, EdgeFashion, NoisyMNIST and NoisyFashion, and a batch size of 128 on RegDB. All experiments are conducted on ten NVIDIA A6000 GPUs (48 GB each).

Our approach employs two variants of the UNet architecture, tailored for different types of image data. Both networks follow the encoder-decoder design paradigm with skip connections but differ in complexity and specific architectural choices to accommodate the characteristics of their respective input data.

### A.6.1 UNET FOR RGB IMAGES

For RGB images such as those from the RegDB dataset, we implement a standard UNet architecture with four encoding and decoding stages. The encoder pathway comprises four blocks, each containing a $3 \times 3$ convolutional layer followed by batch normalization and ReLU activation. Max pooling operations with a $2 \times 2$ kernel progressively reduce spatial dimensions between encoding blocks while increasing feature channels from 64 to 512.

The bottleneck connects the encoder to the decoder through a convolutional layer that expands the feature channels to 1024. The decoder pathway mirrors the encoder with four stages of upsampling. Each decoder block begins with a $2 \times 2$ transposed convolution that doubles the spatial dimensions. Skip connections from corresponding encoder layers are concatenated with the upsampled features to preserve fine-grained spatial information. Each decoder block then applies a $3 \times 3$ convolution with batch normalization and ReLU activation. The network concludes with a $1 \times 1$ convolution mapping to the output channels, followed by a sigmoid activation function.

### A.6.2 UNET_MNIST FOR GRAYSCALE IMAGES

For four grayscale images datasets– EdgeMNIST, NoisyMNIST, EdgeFashion and NoisyFashion– we designed a lightweight variant of UNet with several key modifications. This network employs a shallower architecture with only three encoding stages and significantly fewer parameters. The encoder consists of three convolutional layers with 8, 16, and 32 channels respectively, where downsampling is achieved through strided convolutions (stride=2) rather than separate max pooling operations.

Unlike the standard UNet that uses batch normalization, UNet_MNIST implements instance normalization, which provides better performance for smaller batch sizes and grayscale data. The decoder mirrors the encoder with three upsampling stages using transposed convolutions with stride 2. Each upsampling operation is followed by concatenation with skip connections from the encoder (except for the final layer) and a convolutional block.

Additionally, UNet_MNIST incorporates interpolation operations to ensure proper alignment between feature maps before concatenation, addressing potential size mismatches. The network concludes

Table 5: UNet Architecture for RGB Images (RegDB)

| Layer | Output Dimension | Parameters |
|---|---|---|
| **Encoder** | | |
| Input | $(B, 3, H, W)$ | - |
| Enc1 | $(B, 64, H, W)$ | $3 \times 3$ conv, BatchNorm, ReLU |
| MaxPool | $(B, 64, H/2, W/2)$ | $2 \times 2$ max pooling |
| Enc2 | $(B, 128, H/2, W/2)$ | $3 \times 3$ conv, BatchNorm, ReLU |
| MaxPool | $(B, 128, H/4, W/4)$ | $2 \times 2$ max pooling |
| Enc3 | $(B, 256, H/4, W/4)$ | $3 \times 3$ conv, BatchNorm, ReLU |
| MaxPool | $(B, 256, H/8, W/8)$ | $2 \times 2$ max pooling |
| Enc4 | $(B, 512, H/8, W/8)$ | $3 \times 3$ conv, BatchNorm, ReLU |
| MaxPool | $(B, 512, H/16, W/16)$ | $2 \times 2$ max pooling |
| **Bottleneck** | | |
| Bottleneck | $(B, 1024, H/16, W/16)$ | $3 \times 3$ conv, BatchNorm, ReLU |
| **Decoder** | | |
| UpConv4 | $(B, 512, H/8, W/8)$ | $2 \times 2$ transposed conv |
| Concat | $(B, 1024, H/8, W/8)$ | Concatenate with Enc4 features |
| Dec4 | $(B, 512, H/8, W/8)$ | $3 \times 3$ conv, BatchNorm, ReLU |
| UpConv3 | $(B, 256, H/4, W/4)$ | $2 \times 2$ transposed conv |
| Concat | $(B, 512, H/4, W/4)$ | Concatenate with Enc3 features |
| Dec3 | $(B, 256, H/4, W/4)$ | $3 \times 3$ conv, BatchNorm, ReLU |
| UpConv2 | $(B, 128, H/2, W/2)$ | $2 \times 2$ transposed conv |
| Concat | $(B, 256, H/2, W/2)$ | Concatenate with Enc2 features |
| Dec2 | $(B, 128, H/2, W/2)$ | $3 \times 3$ conv, BatchNorm, ReLU |
| UpConv1 | $(B, 64, H, W)$ | $2 \times 2$ transposed conv |
| Concat | $(B, 128, H, W)$ | Concatenate with Enc1 features |
| Dec1 | $(B, 64, H, W)$ | $3 \times 3$ conv, BatchNorm, ReLU |
| **Output Layer** | | |
| Conv | $(B, 3, H, W)$ | $1 \times 1$ conv |
| Sigmoid | $(B, 3, H, W)$ | Activation function |

with a final transposed convolution that upsamples beyond the original input resolution, followed by a convolutional layer and sigmoid activation. The output is then resized to match the original input dimensions if necessary.

The UNet_MNIST architecture is designed to be computationally efficient while maintaining the ability to capture relevant features from simpler grayscale images, requiring only a fraction of the parameters used in the standard UNet model.

Table 6: UNet_MNIST Architecture for Grayscale Images (EdgeMNIST, NoisyMNIST, EdgeFashion, NoisyFashion)

| Layer | Output Dimension | Parameters |
|---|---|---|
| **Encoder** | | |
| Input | $(B, 1, H, W)$ | - |
| Enc1 | $(B, 8, H, W)$ | $3 \times 3$ conv (stride=1), InstanceNorm, ReLU |
| Enc2 | $(B, 16, H/2, W/2)$ | $3 \times 3$ conv (stride=2), InstanceNorm, ReLU |
| Enc3 | $(B, 32, H/4, W/4)$ | $3 \times 3$ conv (stride=2), InstanceNorm, ReLU |
| **Bottleneck** | | |
| Bottleneck | $(B, 32, H/4, W/4)$ | Optional bottleneck modules |
| **Decoder** | | |
| UpConv3 | $(B, 16, H/2, W/2)$ | $3 \times 3$ transposed conv (stride=2, padding=1) |
| Concat | $(B, 32, H/2, W/2)$ | Concatenate with Enc2 features |
| Dec3 | $(B, 16, H/2, W/2)$ | $3 \times 3$ conv (stride=1), InstanceNorm, ReLU |
| UpConv2 | $(B, 8, H, W)$ | $3 \times 3$ transposed conv (stride=2, padding=1) |
| Concat | $(B, 16, H, W)$ | Concatenate with Enc1 features |
| Dec2 | $(B, 8, H, W)$ | $3 \times 3$ conv (stride=1), InstanceNorm, ReLU |
| UpConv1 | $(B, 8, 2H, 2W)$ | $3 \times 3$ transposed conv (stride=2, padding=1) |
| Dec1 | $(B, 8, 2H, 2W)$ | $3 \times 3$ conv (stride=1), InstanceNorm, ReLU |
| **Output Layer** | | |
| Conv | $(B, 1, 2H, 2W)$ | $1 \times 1$ conv |
| Sigmoid | $(B, 1, 2H, 2W)$ | Activation function |
| Resize | $(B, 1, H, W)$ | Resize to match original input if needed |

## A.7 APPENDICES FOR INCOMPLETE DMVC CLUSTERING PERFORMANCE (SECTION 4.1)

In this section, we provide the clustering of three incomplete DMVC methods with existing rate 0.5 and 0.3 on four datasets, including EdgeFashion, NoisyMNIST, NoisyFashion, and RegDB. The results of EdgeMNIST has been reported in the main paper Table 3.

Table 7: Pre-attack (PRE), post-attack (POST) and post-purification (**Ours**) performance for incomplete DMVC methods on four datasets with existing rate 0.5.

| Method | Phase | EdgeFashion | | | NoisyMNIST | | | NoisyFashion | | | RegDB | | |
|---|---|---|---|---|---|---|---|---|---|---|---|---|---|
| | | ACC | NMI | Purity | ACC | NMI | Purity | ACC | NMI | Purity | ACC | NMI | Purity |
| APADC (TIP'23) | PRE | 0.35 | 0.35 | 0.40 | 0.70 | 0.65 | 0.73 | 0.76 | 0.61 | 0.76 | 0.42 | 0.39 | 0.49 |
| | POST | *0.15* | *0.11* | *0.16* | *0.11* | *0.00* | *0.11* | *0.10* | *0.00* | *0.10* | *0.26* | *0.19* | *0.26* |
| | **Ours** | 0.36 | 0.29 | 0.39 | 0.45 | 0.38 | 0.47 | 0.31 | 0.29 | 0.35 | 0.36 | 0.27 | 0.40 |
| DVIMC (AAAI'24) | PRE | 0.54 | 0.56 | 0.56 | 0.84 | 0.79 | 0.84 | 0.69 | 0.66 | 0.70 | 0.63 | 0.68 | 0.66 |
| | POST | *0.27* | *0.19* | *0.29* | *0.22* | *0.12* | *0.25* | *0.34* | *0.26* | *0.28* | *0.34* | *0.43* | *0.40* |
| | **Ours** | 0.50 | 0.51 | 0.53 | 0.32 | 0.21 | 0.33 | 0.42 | 0.36 | 0.42 | 0.37 | 0.45 | 0.42 |
| LOGIC (NN'25) | PRE | 0.53 | 0.53 | 0.57 | 0.45 | 0.32 | 0.45 | 0.41 | 0.37 | 0.42 | 0.33 | 0.32 | 0.37 |
| | POST | *0.27* | *0.20* | *0.27* | *0.34* | *0.23* | *0.35* | *0.34* | *0.32* | *0.36* | *0.33* | *0.32* | *0.37* |
| | **Ours** | 0.40 | 0.35 | 0.41 | 0.44 | 0.36 | 0.47 | 0.40 | 0.35 | 0.42 | 0.34 | 0.34 | 0.38 |

## A.8 APPENDICES FOR CROSS-DATASETS PURIFICATION ( SECTION 4.3)

In this section, we supplement the main paper with additional results (NMI and Purity) on the transfer matrix of EAMC for purifiers. The transfer matrix is a useful tool to evaluate the performance of the

Table 8: Pre-attack (PRE), post-attack (POST) and post-purification (**Ours**) performance for incomplete DMVC methods on four datasets with existing rate 0.3.

| Method | Phase | EdgeFashion | | | NoisyMNIST | | | NoisyFashion | | | RegDB | | |
|---|---|---|---|---|---|---|---|---|---|---|---|---|---|
| | | ACC | NMI | Purity | ACC | NMI | Purity | ACC | NMI | Purity | ACC | NMI | Purity |
| APADC (TIP'23) | PRE | 0.34 | 0.31 | 0.37 | 0.71 | 0.64 | 0.75 | 0.67 | 0.58 | 0.69 | 0.41 | 0.37 | 0.46 |
| | POST | *0.17* | *0.09* | *0.18* | *0.11* | *0.00* | *0.12* | *0.11* | *0.00* | *0.11* | *0.35* | *0.24* | *0.37* |
| | **Ours** | 0.32 | 0.28 | 0.36 | 0.27 | 0.19 | 0.28 | 0.30 | 0.26 | 0.34 | 0.45 | 0.38 | 0.48 |
| DVIMC (AAAI'24) | PRE | 0.54 | 0.57 | 0.58 | 0.68 | 0.71 | 0.73 | 0.64 | 0.64 | 0.67 | 0.51 | 0.60 | 0.56 |
| | POST | *0.29* | *0.26* | *0.32* | *0.34* | *0.23* | *0.35* | *0.29* | *0.21* | *0.29* | *0.36* | *0.47* | *0.44* |
| | **Ours** | 0.49 | 0.51 | 0.53 | 0.42 | 0.31 | 0.44 | 0.42 | 0.38 | 0.43 | 0.43 | 0.53 | 0.47 |
| LOGIC (NN'25) | PRE | 0.45 | 0.47 | 0.51 | 0.35 | 0.28 | 0.36 | 0.35 | 0.32 | 0.37 | 0.34 | 0.38 | 0.38 |
| | POST | *0.19* | *0.14* | *0.19* | *0.27* | *0.19* | *0.29* | *0.28* | *0.25* | *0.31* | *0.26* | *0.27* | *0.30* |
| | **Ours** | 0.38 | 0.31 | 0.38 | 0.31 | 0.24 | 0.32 | 0.38 | 0.32 | 0.39 | 0.30 | 0.31 | 0.34 |

purifier when trained on one dataset and tested on another. The results are shown in Figure 6a and Figure 6b. Each column denotes the source dataset used to train the purifier, and each row denotes the target dataset on which NMI and Purity are evaluated.

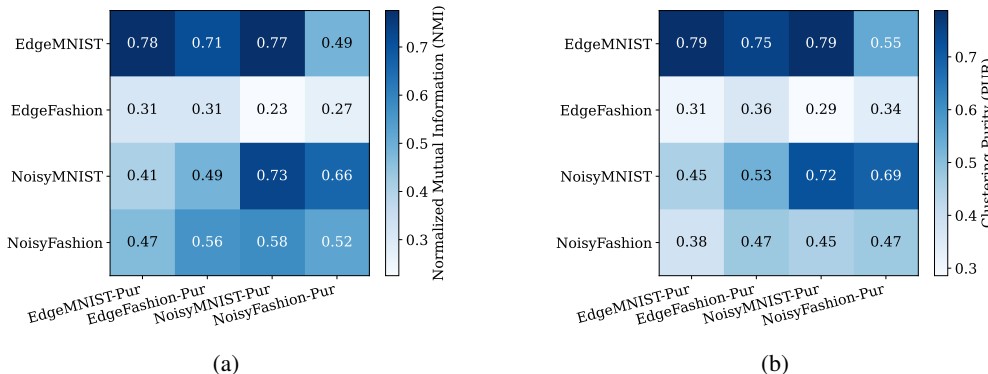

(a)                                                           (b)

Figure 6: (a) and (b): Transfer matrix of EAMC for purifiers: each column denotes the source dataset used to train the purifier, and each row denotes the target dataset on which NMI and Purity are evaluated.

### A.9 APPENDICES FOR CROSS-MODELS PURIFICATION

This section augments the main paper by presenting supplementary NMI and Purity results for the EAMC method, derived from a cross-model purifier transfer matrix analysis (Figures 7a and 7b). The transfer matrix framework is a valuable instrument for evaluating purifier generalization, specifically by assessing performance when a purifier trained on a source model is subsequently applied to a distinct target model. In the depicted figures, each column denotes the source model utilized for purifier training, while each row indicates the target model on which NMI and Purity metrics were evaluated.

### A.10 APPENDICES FOR ATTACK STRENGTHS ( SECTION 4.5)

In this section, we supplement the main paper with additional results on the attack strengths of different methods. The results are shown in Figures 8, 9, 10 and 11.

### A.11 SENSITIVITY ANALYSIS OF WEIGHTING HYPERPARAMETERS

A key aspect of our proposed framework is the interplay of the three loss components, weighted by the hyperparameters $\alpha$, $\beta$, and $\gamma$ in the overall objective function (Eq. 8). To assess the stability of our method with respect to these parameters, we conducted a sensitivity analysis. We performed

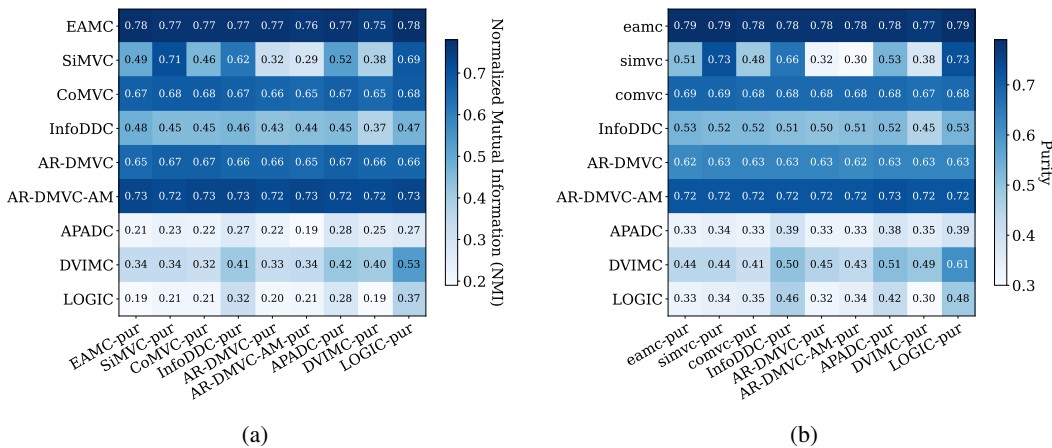

(a)                                                                 (b)

Figure 7: (a) and (b): Transfer matrix of NMI and Purity for purifiers: each column denotes the source model used to train the purifier, and each row denotes the target model.

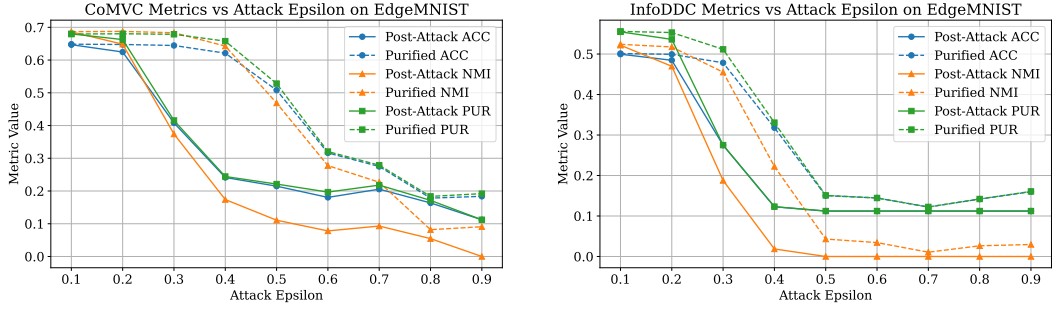

Figure 8: CoMVC and InfoDDC vs. attack strengths on EdgeMNIST

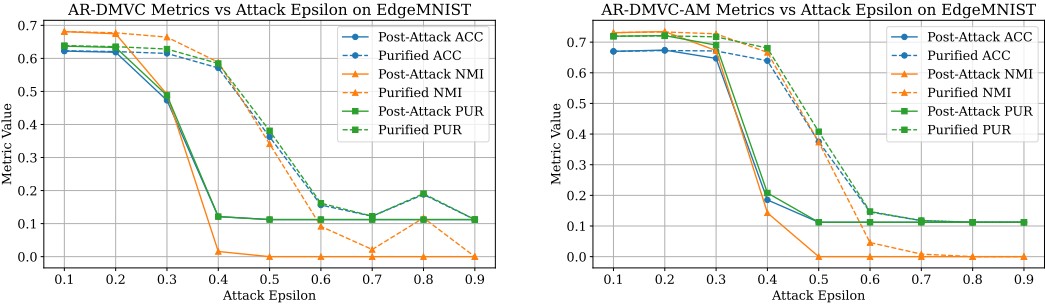

Figure 9: AR-DMVC and AR-DMVC-AM vs. attack strengths on EdgeMNIST

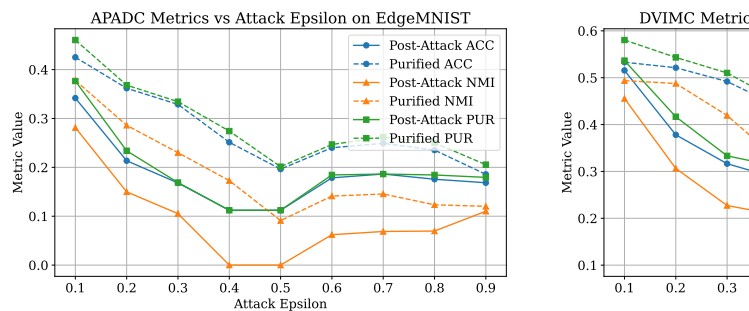

Figure 10: APADC and DVIMC vs. attack strengths on EdgeMNIST

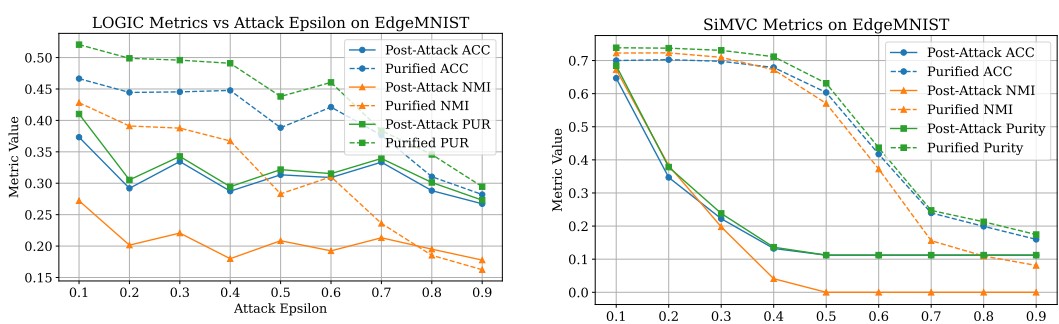

Figure 11: LOGIC and SiMVC vs. attack strengths on EdgeMNIST

the analysis on the SiMVC model using the EdgeMNIST dataset. For each parameter ($\alpha$, $\beta$, $\gamma$), we varied its value across a wide range from 0.01 to 100, while keeping the other two parameters fixed at a default value of 1. The post-purification clustering performance (ACC, NMI, and Purity) is reported in Table 9.

Table 9: Sensitivity analysis of the weighting hyperparameters ($\alpha, \beta, \gamma$) on the SiMVC model for the EdgeMNIST dataset. For each experiment, one parameter is varied while the other two are held constant at 1. Performance is measured in post-purification ACC, NMI, and Purity.

| Metric | Varying $\alpha$ | | | | | Varying $\beta$ | | | | | Varying $\gamma$ | | | | |
|---|---|---|---|---|---|---|---|---|---|---|---|---|---|---|---|
| | 0.01 | 0.1 | 1 | 10 | 100 | 0.01 | 0.1 | 1 | 10 | 100 | 0.01 | 0.1 | 1 | 10 | 100 |
| ACC | 0.71 | 0.71 | 0.71 | 0.71 | 0.70 | 0.69 | 0.71 | 0.71 | 0.70 | 0.70 | 0.71 | 0.71 | 0.71 | 0.70 | 0.70 |
| NMI | 0.73 | 0.73 | 0.73 | 0.73 | 0.72 | 0.69 | 0.72 | 0.73 | 0.70 | 0.70 | 0.73 | 0.72 | 0.73 | 0.68 | 0.68 |
| Purity | 0.74 | 0.74 | 0.74 | 0.74 | 0.74 | 0.72 | 0.74 | 0.74 | 0.73 | 0.73 | 0.74 | 0.74 | 0.74 | 0.71 | 0.71 |

The results demonstrate that our framework is highly robust to the choice of these hyperparameters. For all three parameters, performance remains remarkably stable across several orders of magnitude. For instance, varying $\alpha$ from 0.01 to 100 results in a negligible change in performance. While very large values of $\gamma$ (e.g., 10 or 100) cause a minor drop in NMI and Purity, the ACC score remains stable.

Given this observed stability, a systematic guideline for setting these parameters is straightforward. We recommend setting $\alpha = \beta = \gamma = 1$ as a simple and effective default. This configuration requires no exhaustive tuning and, as shown in our experiments, yields consistently strong performance. We used this default setting for the experiments in the main paper unless otherwise specified.

### A.12 Performance Under Extreme Missing Data Scenarios

To further probe the robustness of our framework, we conducted an analysis under extreme missing data scenarios, pushing the limits of data availability beyond the experiments presented in the main paper ( Table 3). This experiment is designed to evaluate whether our purification step remains beneficial when the underlying DMVC models operate with minimal-to-no overlapping view information.

We focused on the three incomplete DMVC methods evaluated in our paper, using the EdgeMNIST dataset. The missing rates were set to near-total scarcity: For **APADC** and **LOGIC**, which are two-view methods, we set the view existence rate to 0%. This means every sample in the dataset had only one of its two views present, with no single sample possessing both views. For **DVIMC**, which requires a small number of complete multi-view samples to function, we used a minimal 1% existence rate.

The results of this extreme evaluation are presented in Table 10. As expected, the baseline pre-attack (PRE) performance for all methods is significantly lower than in less severe settings, as the models themselves struggle to perform effective clustering with such scarce information. Consequently, the absolute performance degradation from an adversarial attack (POST) is also limited.

Table 10: Performance of incomplete DMVC methods under extreme missing rates on the EdgeMNIST dataset. The existence rate for APADC and LOGIC is 0%; for DVIMC, it is 1%.

| Method | Phase | ACC | NMI | Purity |
|--------|-------|-----|-----|--------|
| **APADC** | PRE | 0.24 | 0.17 | 0.28 |
| | POST | 0.21 | 0.12 | 0.23 |
| | **Ours** | **0.24** | **0.16** | **0.25** |
| **DVIMC** | PRE | 0.39 | 0.32 | 0.40 |
| | POST | 0.25 | 0.17 | 0.26 |
| | **Ours** | **0.27** | **0.18** | **0.30** |
| **LOGIC** | PRE | 0.28 | 0.24 | 0.33 |
| | POST | 0.21 | 0.12 | 0.23 |
| | **Ours** | **0.31** | **0.21** | **0.31** |

Despite these challenging conditions, our purification step (**Ours**) consistently improves the post-attack performance across all metrics for all three methods. Notably, for LOGIC, our method not only recovers the performance but slightly surpasses its original pre-attack accuracy (0.31 vs. 0.28 ). This analysis demonstrates that even at the absolute limit of data availability, our framework remains competitive and beneficial, providing measurable performance gains and underscoring the robustness of our purification strategy.

### A.13 Appendices for Different $\mathcal{D}$

Overall, Euclidean distance yields small positive deviations (up to +0.03) over Cosine in both NMI and Purity for most methods, indicating a slight edge in clustering quality. In contrast, Wasserstein distance induces substantial negative drops (as large as –0.38 in Purity for SIMVC), reflecting its sensitivity to perturbations. These patterns confirm that Cosine and Euclidean metrics are comparably robust, whereas Wasserstein exhibits pronounced instability under adversarial noise.

### A.14 Evaluation Against White-Box Attacks

The main experiments in our paper focus on the black-box attack specifically designed for DMVC by Huang et al. (2024), as the field of adversarial attacks for this task is still nascent. To further broaden our evaluation and test the resilience of GUARD, we also conducted experiments against the classic white-box Fast Gradient Sign Method (FGSM) attack (Goodfellow et al., 2014).

Adapting white-box attacks to the unsupervised, multi-view clustering setting presents non-trivial challenges. For this analysis, we focused on several representative complete DMVC methods, where

Table 11: NMI of different distance metrics on EdgeMNIST with $\epsilon = 0.3$, using Cosine as baseline.

| Method | Cosine | Euclidean | Wasserstein |
|---|---|---|---|
| EAMC | 0.77 | 0.78 (+0.01) | 0.77 (-0.00) |
| SIMVC | 0.71 | 0.74 (+0.03) | 0.35 (-0.36) |
| COMVC | 0.68 | 0.69 (+0.01) | 0.68 (-0.01) |
| MIMVC | 0.47 | 0.49 (+0.03) | 0.41 (-0.05) |
| AR-DMVC | 0.66 | 0.67 (+0.01) | 0.54 (-0.12) |
| AR-DMVC-AM | 0.73 | 0.66 (-0.07) | 0.73 (+0.00) |
| APADC | 0.26 | 0.23 (-0.03) | 0.27 (+0.01) |
| DVIMC | 0.38 | 0.40 (+0.02) | 0.29 (-0.10) |
| LOGIC | 0.37 | 0.39 (+0.02) | 0.32 (-0.05) |

Table 12: Purity of different distance metrics on EdgeMNIST with $\epsilon = 0.3$, using Cosine as baseline.

| Method | Cosine | Euclidean | Wasserstein |
|---|---|---|---|
| EAMC | 0.78 | 0.78 (+0.00) | 0.78 (-0.00) |
| SIMVC | 0.73 | 0.75 (+0.02) | 0.35 (-0.38) |
| COMVC | 0.68 | 0.68 (+0.00) | 0.68 (-0.00) |
| MIMVC | 0.53 | 0.54 (+0.01) | 0.49 (-0.03) |
| AR-DMVC | 0.63 | 0.63 (+0.00) | 0.53 (-0.10) |
| AR-DMVC-AM | 0.72 | 0.68 (-0.04) | 0.72 (+0.00) |
| APADC | 0.41 | 0.34 (-0.07) | 0.39 (-0.02) |
| DVIMC | 0.48 | 0.51 (+0.03) | 0.41 (-0.07) |
| LOGIC | 0.48 | 0.50 (+0.01) | 0.42 (-0.07) |

model gradients required for a white-box attack are more directly accessible. We note that other classic attacks like C&W (Carlini & Wagner, 2017) are not readily applicable, as they require classification logits that are unavailable in our fully unsupervised setting.

The results on the EdgeMNIST dataset are presented in Table 13. The table shows the performance of various DMVC methods after being attacked by FGSM (POST) and after being subsequently purified by our GUARD framework (**PUR**).

Table 13: Performance restoration against the white-box FGSM attack on the EdgeMNIST dataset ($\epsilon = 0.3$). Our purification step (**Ours**) provides a substantial performance improvement over the post-attack (POST) state for all tested methods.

| Method | Phase | ACC | NMI | Purity |
|---|---|---|---|---|
| **EAMC** | POST | 0.53 | 0.47 | 0.56 |
| | **Ours** | **0.71** | **0.71** | **0.75** |
| **SiMVC** | POST | 0.28 | 0.18 | 0.30 |
| | **Ours** | **0.63** | **0.57** | **0.65** |
| **CoMVC** | POST | 0.35 | 0.25 | 0.35 |
| | **Ours** | **0.59** | **0.56** | **0.62** |
| **InfoDDC** | POST | 0.31 | 0.21 | 0.34 |
| | **Ours** | **0.40** | **0.35** | **0.46** |
| **AR-DMVC** | POST | 0.60 | 0.60 | 0.65 |
| | **Ours** | **0.63** | **0.64** | **0.68** |

The results clearly demonstrate that GUARD provides substantial performance restoration against the white-box FGSM attack on all tested complete DMVC models. This additional experiment validates the versatility of our purification framework against different threat models. We note that adapting

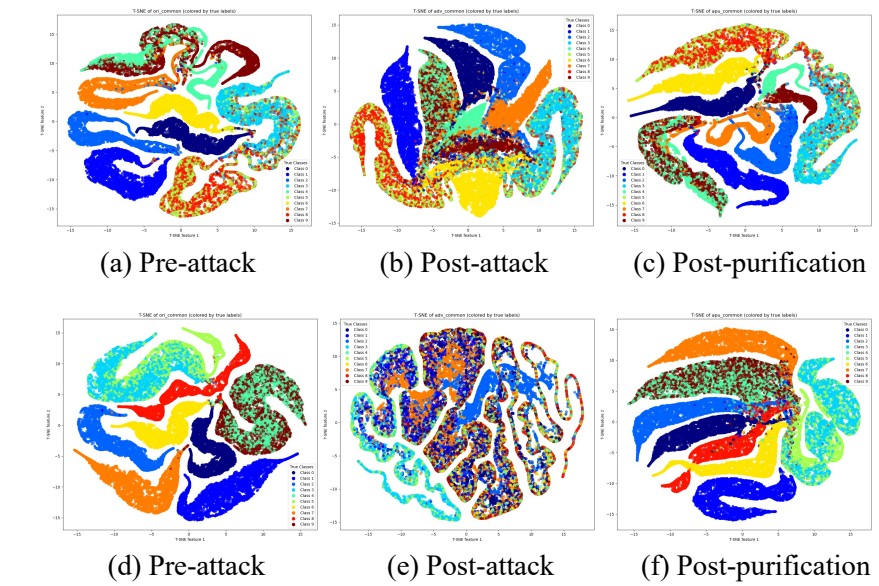

Figure 12: The T-SNE visualization of the EAMC on the EdgeMNIST (a-c) and NoisyMNIST (d-f).

white-box attacks to the more complex architectures of *incomplete* DMVC methods presents unique implementation challenges, which we leave as an interesting direction for future work.

### A.15 T-SNE VISUALIZATION

From the T-SNE visualization in Figure 12, we observe that the clusters of the Pre-attack are well separated, while the clusters of the Post-attack are significantly mixed. After purification, the clusters of the Post-purification become more compact and distinct as the Pre-attack, indicating that our proposed GUARD effectively purifies the adversarial perturbations and enhances the clustering performance.

### A.16 VISUALIZATIONS OF THE PURIFIED IMAGES

In this section, we supplement the main paper with additional visualizations of the purified images. The visualizations are provided for all methods, including EAMC, SiMVC, CoMVC, InfoDDC, AR-DMVC, AR-DMVC-AM, APADC, DVIMC and LOGIC. The images are presented in three columns: Pre-attack (Clean), Post-attack (Attack) and Post-purification (Purified, ours). The visualizations are shown in Figures 13 to 30.

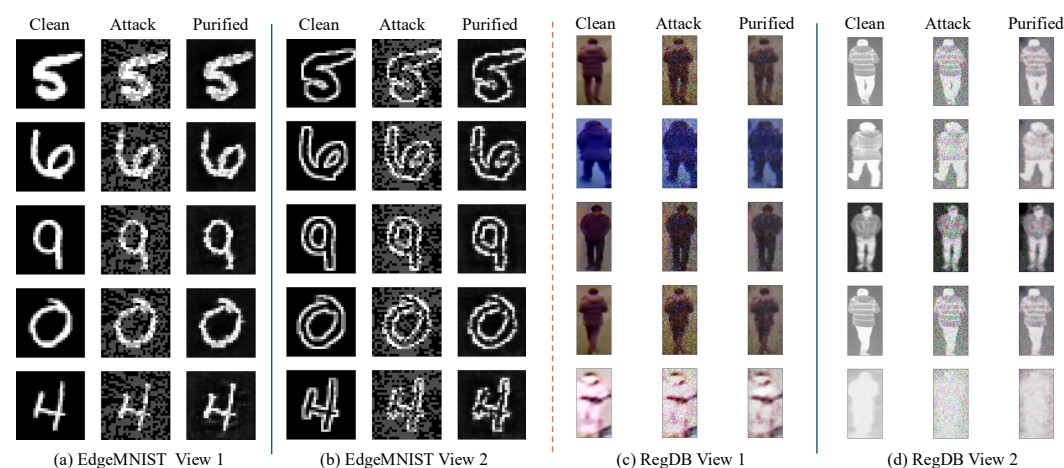

Figure 13: A visualization of the Pre-attack (Clean), Post-attack (Attack) and Post-purification (Purified, ours) images for EAMC in EdgeMNIST and RegDB.

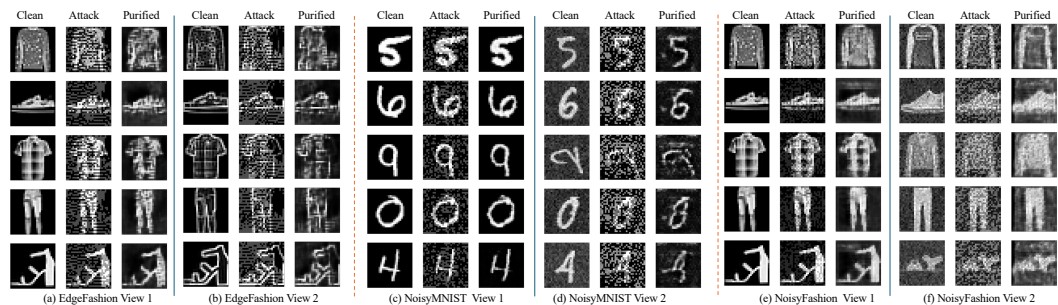

Figure 14: A visualization of the Pre-attack (Clean), Post-attack (Attack) and Post-purification (Purified, ours) images for EAMC in EdgeFashion, NoisyMNIST and NoisyFashion.

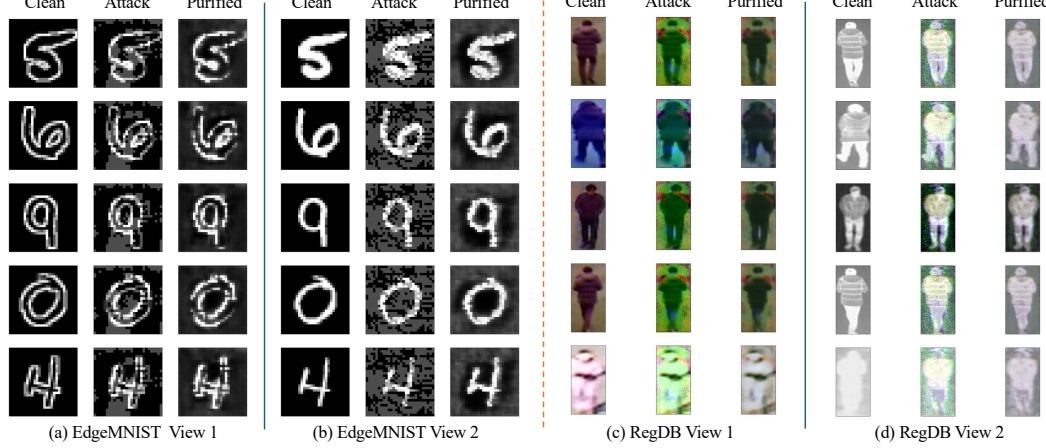

Figure 15: A visualization of the Pre-attack (Clean), Post-attack (Attack) and Post-purification (Purified, ours) images for SiMVC in EdgeMNIST and RegDB.

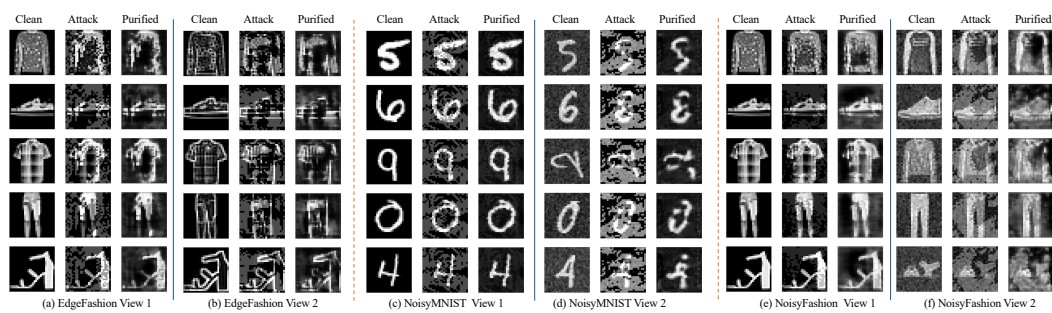

Figure 16: A visualization of the Pre-attack (Clean), Post-attack (Attack) and Post-purification (Purified, ours) images for SiMVC in EdgeFashion, NoisyMNIST and NoisyFashion.

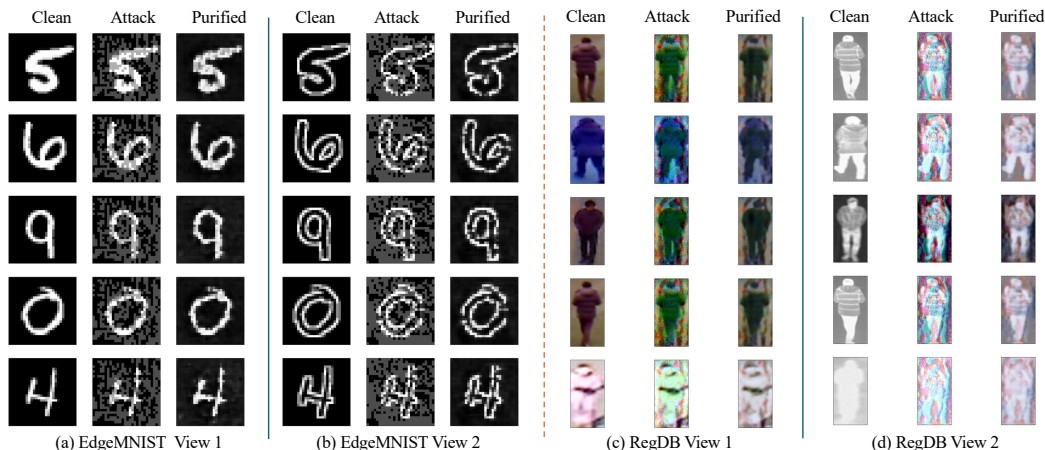

Figure 17: A visualization of the Pre-attack (Clean), Post-attack (Attack) and Post-purification (Purified, ours) images for CoMVC in EdgeMNIST and RegDB.

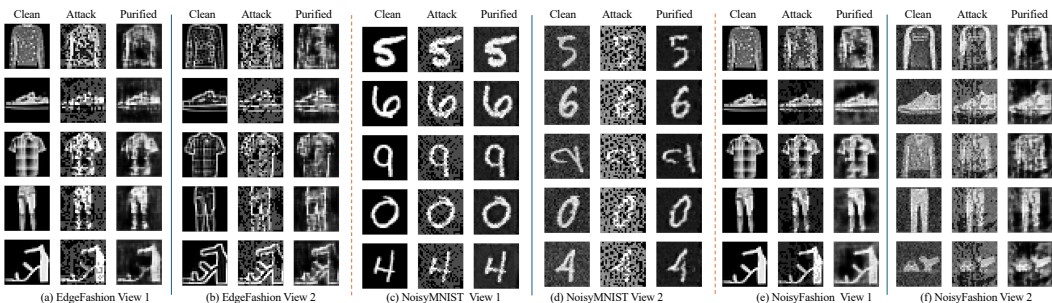

Figure 18: A visualization of the Pre-attack (Clean), Post-attack (Attack) and Post-purification (Purified, ours) images for CoMVC in EdgeFashion, NoisyMNIST and NoisyFashion.

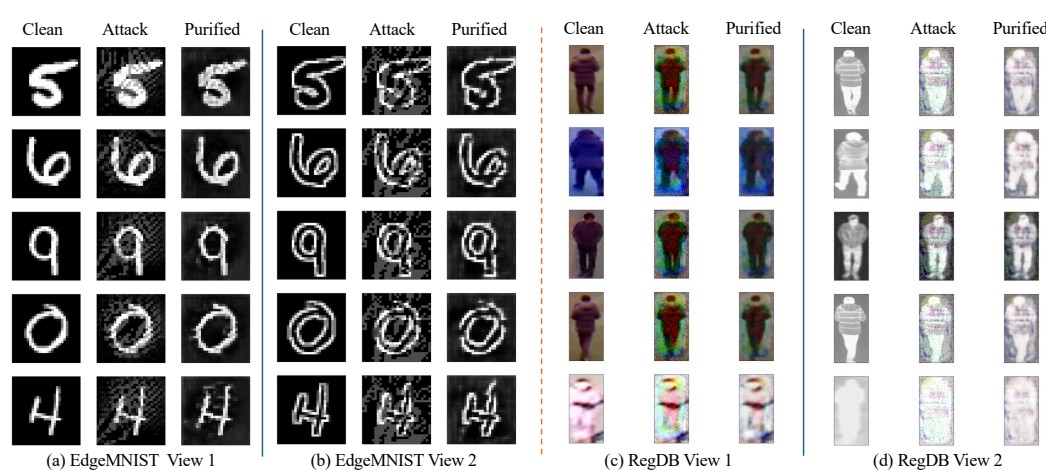

(a) EdgeMNIST View 1    (b) EdgeMNIST View 2    (c) RegDB View 1    (d) RegDB View 2

Figure 19: A visualization of the Pre-attack (Clean), Post-attack (Attack) and Post-purification (Purified, ours) images for InfoDDC in EdgeMNIST and RegDB.

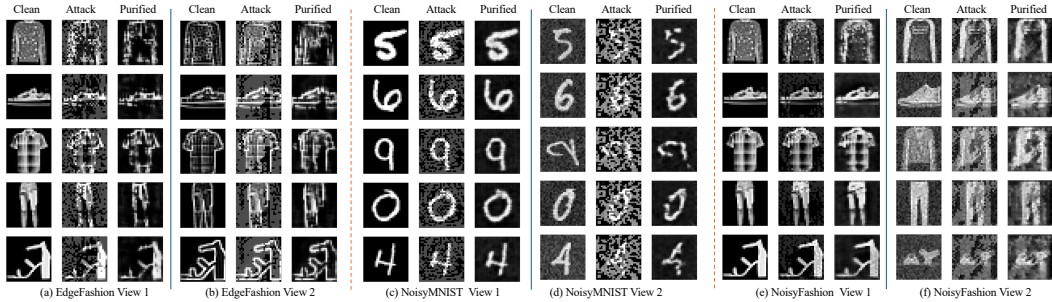

(a) EdgeFashion View 1    (b) EdgeFashion View 2    (c) NoisyMNIST View 1    (d) NoisyMNIST View 2    (e) NoisyFashion View 1    (f) NoisyFashion View 2

Figure 20: A visualization of the Pre-attack (Clean), Post-attack (Attack) and Post-purification (Purified, ours) images for InfoDDC in EdgeFashion, NoisyMNIST and NoisyFashion.

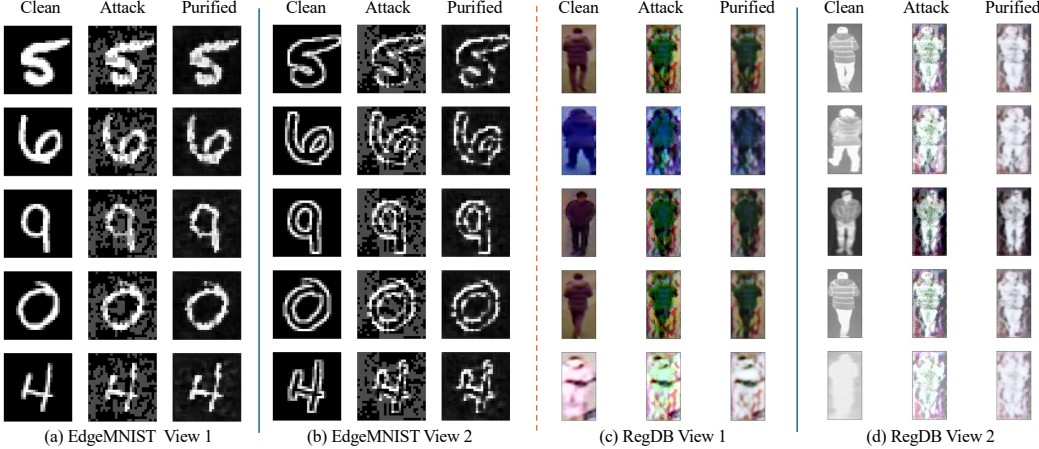

(a) EdgeMNIST View 1    (b) EdgeMNIST View 2    (c) RegDB View 1    (d) RegDB View 2

Figure 21: A visualization of the Pre-attack (Clean), Post-attack (Attack) and Post-purification (Purified, ours) images for AR-DMVC in EdgeMNIST and RegDB.

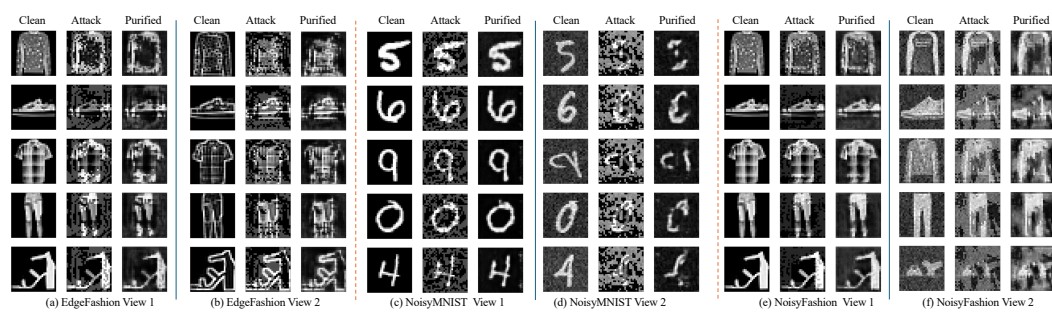

Figure 22: A visualization of the Pre-attack (Clean), Post-attack (Attack) and Post-purification (Purified, ours) images for AR-DMVC in EdgeFashion, NoisyMNIST and NoisyFashion.

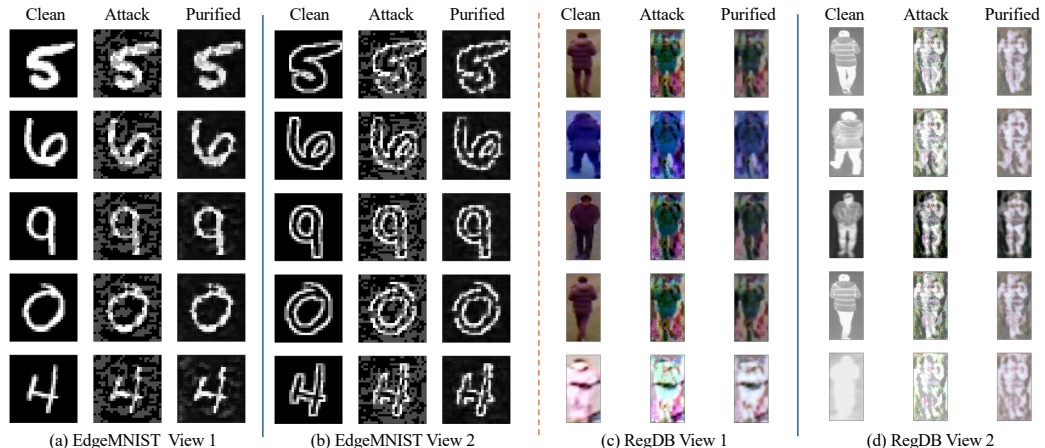

Figure 23: A visualization of the Pre-attack (Clean), Post-attack (Attack) and Post-purification (Purified, ours) images for AR-DMVC-AM in EdgeMNIST and RegDB.

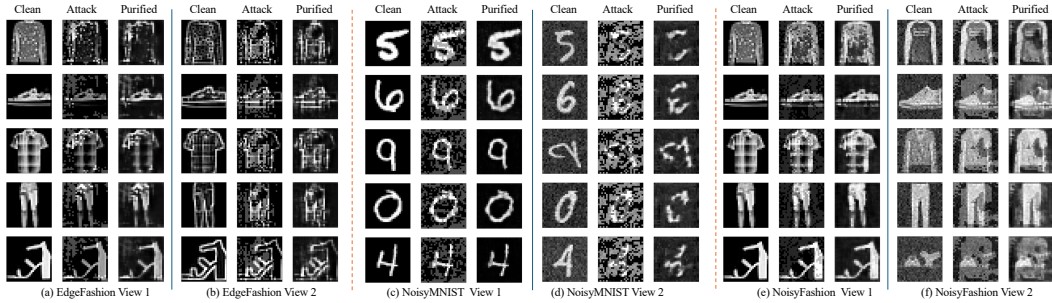

Figure 24: A visualization of the Pre-attack (Clean), Post-attack (Attack) and Post-purification (Purified, ours) images for AR-DMVC-AM in EdgeFashion, NoisyMNIST and NoisyFashion.

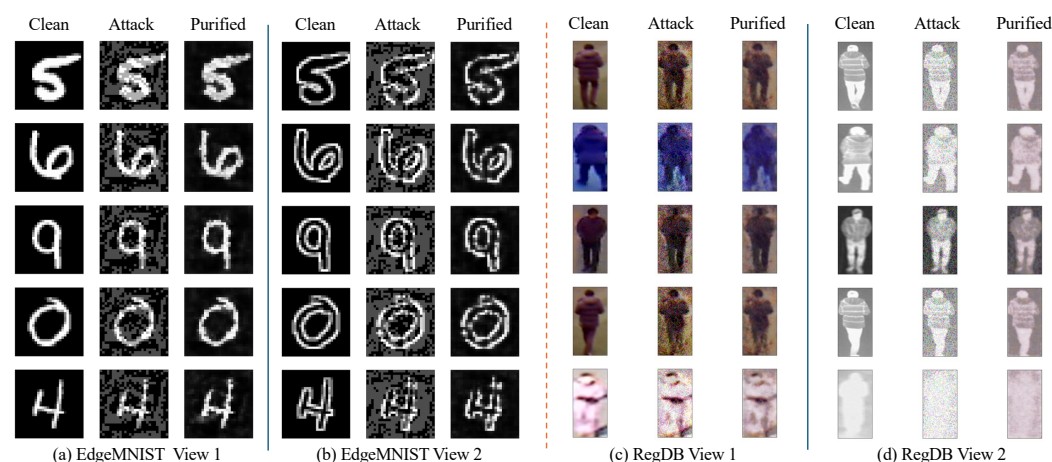

Figure 25: A visualization of the Pre-attack (Clean), Post-attack (Attack) and Post-purification (Purified, ours) images for APADC in EdgeMNIST and RegDB.

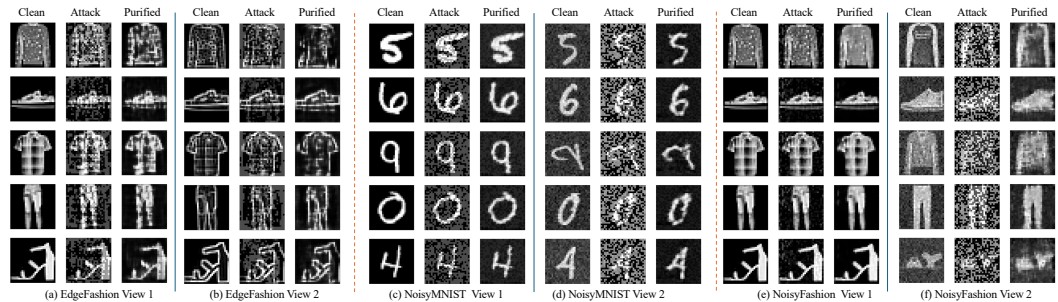

Figure 26: A visualization of the Pre-attack (Clean), Post-attack (Attack) and Post-purification (Purified, ours) images for APADC in EdgeFashion, NoisyMNIST and NoisyFashion.

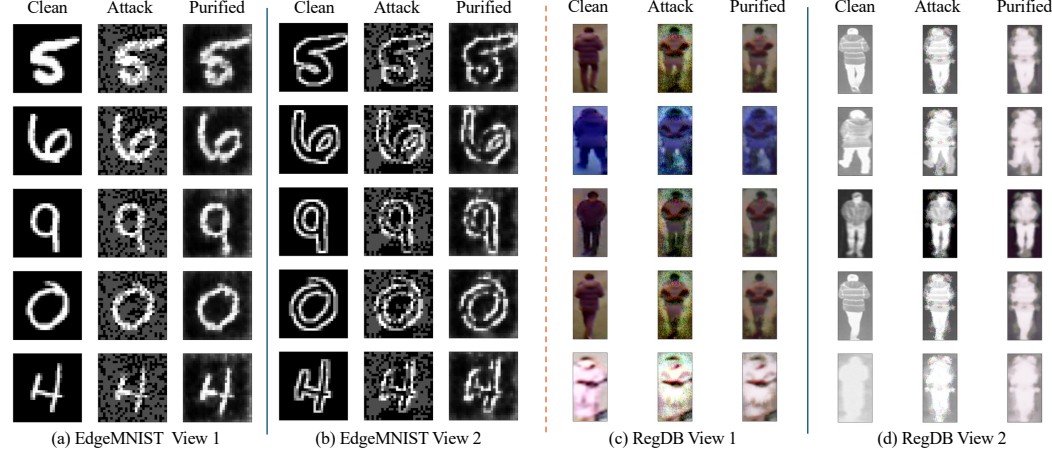

Figure 27: A visualization of the Pre-attack (Clean), Post-attack (Attack) and Post-purification (Purified, ours) images for DVIMC in EdgeMNIST and RegDB.

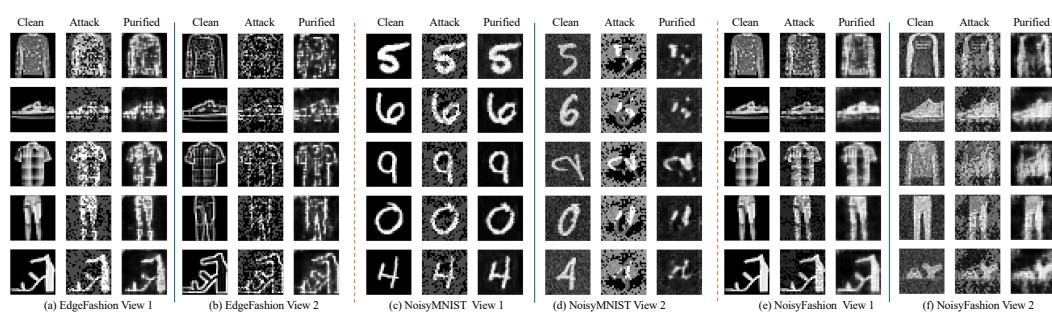

Figure 28: A visualization of the Pre-attack (Clean), Post-attack (Attack) and Post-purification (Purified, ours) images for DVIMC in EdgeFashion, NoisyMNIST and NoisyFashion.

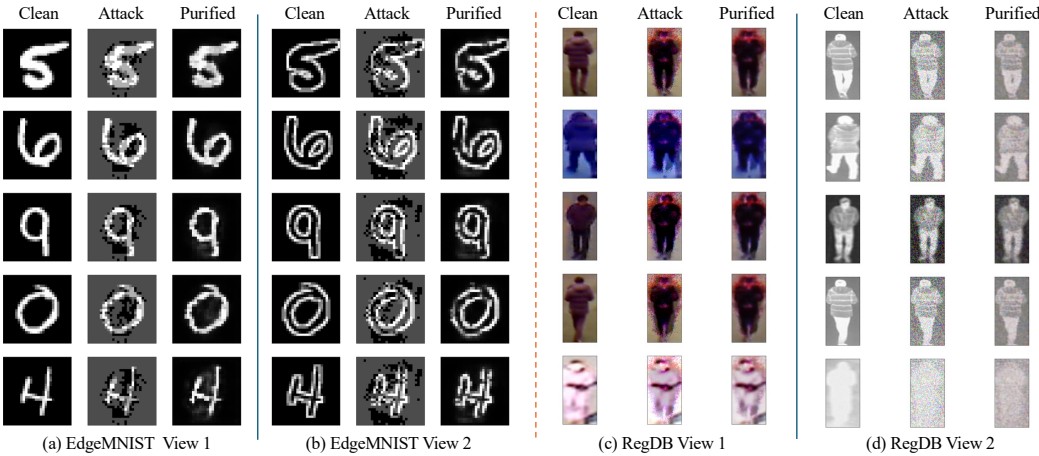

Figure 29: A visualization of the Pre-attack (Clean), Post-attack (Attack) and Post-purification (Purified, ours) images for LOGIC in EdgeMNIST and RegDB.

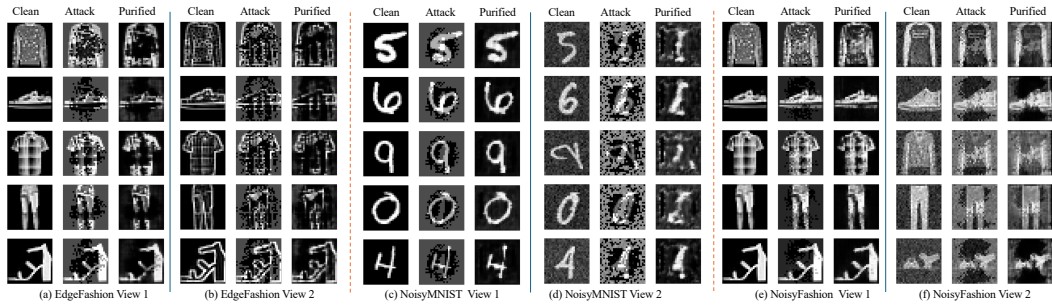

Figure 30: A visualization of the Pre-attack (Clean), Post-attack (Attack) and Post-purification (Purified, ours) images for LOGIC in EdgeFashion, NoisyMNIST and NoisyFashion.

