# OpenReview forum: "GUARD: General Unsupervised Adversarial Robust Defense for Deep Multi-View Clustering via Information Bottleneck"
_ICLR.cc/2026/Conference — ICLR 2026 Conference Withdrawn Submission_

### Official Review · Reviewer_FEdh · 2025-10-19

**Soundness:** 2
**Presentation:** 3
**Contribution:** 2
**Rating:** 4
**Confidence:** 3

**Summary:**

This article proposes an unsupervised defense framework called GUARD based on the "information bottleneck". By "purifying" the malicious perturbation signals in the input data, it significantly enhances the robustness of multi-view clustering models against adversarial attacks, while also maintaining high efficiency and generalization capabilities.

**Strengths:**

1. The proposed GUARD purifier does not rely on a specific clustering model structure. It can seamlessly integrate with various DMVC methods, and demonstrates certain transferability in experiments across datasets and models.
2. This paper for the first time models multi-view adversarial defense as an "information separation" problem. By leveraging the Multi-view Information Bottleneck (MIB) principle, it integrates "preserving task-related information" and "removing adversarial noise" into a unified theoretical framework, demonstrating high innovation and theoretical value.

**Weaknesses:**

1. Although the paper claims to be an unsupervised method, the purifier needs to use the original clean data as a supervisory signal to align the output during training. This is often difficult to obtain in real attack scenarios, which limits the method's usability in practical applications.
2. Could the authors further explore whether their approach can be extended to other tasks, such as classification or detection, or evaluated on larger and more complex datasets?

**Questions:**

1. GUARD needs supervision signals based on clean data, but this is somewhat contradictory in an unsupervised scenario. The core of the loss function is to use clean data to supervise the output. In real attack scenarios, it may not be possible to obtain completely clean data or only partial data, which will affect the practical usability of the method.
2. Although the paper claims that it is approximately 400 times faster than diffusion purification. How will it perform in large-scale and high-resolution tasks?
3. If the network capacity is large, or if the noise and data are strongly correlated, could the model possibly retain some of the noise instead of being "naturally" discarded?
4. Does the author fail to include "adaptive attack"? If the attacker is aware that the GUARD purifier exists, will the GUARD perform well?

---

### Official Review · Reviewer_oqXW · 2025-10-27

**Soundness:** 3
**Presentation:** 3
**Contribution:** 3
**Rating:** 6
**Confidence:** 4

**Summary:**

This paper presents GUARD, a novel multi-view adversarial purification framework that introduces the new paradigm of Multi-view Adversarial Purification (MAP). The authors reformulate adversarial defense as an information-theoretic signal separation problem. Specifically, GUARD trains a plug-and-play purifier based on the information bottleneck principle to extract task-relevant clean signals from adversarially perturbed inputs while filtering out adversarial noise.

A key innovation lies in its self-supervised training mechanism, which focuses solely on information sufficiency—by enforcing consistency between purified and clean data at both pixel and feature levels, “purity” naturally emerges during optimization without any explicit regularization. Overall, the paper is technically sound, well-motivated, and supported by comprehensive experiments demonstrating strong model-agnostic performance, cross-dataset generalization, and high computational efficiency.

**Strengths:**

1. The proposed method is highly portable and practical. It functions as an independent front-end module that requires no dataset- or model-specific parameter tuning. The purifier generalizes well across datasets and significantly enhances adversarial robustness.
2. Experiments show that GUARD restores attacked models to nearly their clean accuracy while achieving up to 400× faster inference than diffusion-based purification baselines, maintaining comparable or better performance. This makes it highly suitable for real-world deployment.
3. The paper is overall well-executed, with rigorous method design and comprehensive experiments that convincingly demonstrate the effectiveness and efficiency of the proposed approach.

**Weaknesses:**

1. Although the purifier exhibits strong generalization ability, the experiments are limited to image-based multi-view datasets and relatively old downstream models. Expanding evaluations to more diverse datasets and new downstream models would strengthen the paper.
2. This article lacks a detailed complexity analysis of the purifier. Please provide a complete supplement.
3. At present, it mainly targets PGD attacks. Additional robustness analysis can be conducted under other perturbation attacks to verify the stability and effectiveness of the method.

**Questions:**

1. Although the purifier exhibits strong generalization ability, the experiments are limited to image-based multi-view datasets and relatively old downstream models. Expanding evaluations to more diverse datasets and new downstream models would strengthen the paper.
2. This article lacks a detailed complexity analysis of the purifier. Please provide a complete supplement.
3. At present, it mainly targets PGD attacks. Additional robustness analysis can be conducted under other perturbation attacks to verify the stability and effectiveness of the method.

---

### Official Review · Reviewer_mwxm · 2025-10-30

**Soundness:** 2
**Presentation:** 2
**Contribution:** 2
**Rating:** 2
**Confidence:** 4

**Summary:**

This paper proposes GUARD, an unsupervised adversarial defense method for deep multi-view clustering (DMVC). The authors reframe adversarial purification as a signal separation problem through an information-theoretic lens, inspired by the Multi-View Information Bottleneck (MIB) principle. GUARD operationalizes this concept via a self-supervised loss designed to enforce both informational sufficiency and purity without explicit adversarial regularization. The method is evaluated across several DMVC models (e.g., EAMC, SiMVC, InfoDDC, AR-DMVC) and datasets (EdgeMNIST, NoisyMNIST, RegDB), showing improved adversarial robustness over baselines, including diffusion-based purifiers.

**Strengths:**

1. The paper is clearly written and well-structured, with extensive experimental results.

2. The empirical section is thorough, covering both complete and incomplete multi-view clustering.

**Weaknesses:**

1. The concept of enforcing information sufficiency through the information bottleneck for achieving multi-view robustness has already been explored in prior works such as [1] and [2].

2. The theoretical treatment of information sufficiency and mutual information estimation closely mirrors that of Federici et al., The “purification” component is only superficially motivated, being described primarily as adding and mitigating adversarial noise, without deeper analytical justification.

3. It remains unclear why the authors reuse the same mutual information estimation strategy as Federici et al. since directly adopting it does not constitute a theoretical advance. Moreover, the rationale for not directly minimizing I(X, Z) is not discussed or justified.


[1] Yu, Xi, et al. "Improving adversarial robustness by learning shared information." Pattern Recognition 134 (2023): 109054.

[2] Zhang, Qi, et al. "Multi-view information bottleneck without variational approximation." ICASSP 2022-2022 IEEE International Conference on Acoustics, Speech and Signal Processing (ICASSP). IEEE, 2022.

**Questions:**

1. The paper claims that “purity emerges implicitly from the optimization process.” Can the authors provide theoretical justification or empirical evidence supporting this assertion? Additionally, would incorporating an explicit information bottleneck loss further improve or stabilize the purification process?

I would be willing to raise my score if the authors can adequately address my concerns regarding the novelty of the work and the questions raised above

---

### Official Review · Reviewer_onzn · 2025-11-01

**Soundness:** 3
**Presentation:** 3
**Contribution:** 2
**Rating:** 2
**Confidence:** 3

**Summary:**

This paper tackles the vulnerability of deep multi-view clustering (DMVC) models to adversarial attacks by proposing GUARD (General Unsupervised Adversarial Robust Defense), a novel unsupervised and model-agnostic defense framework. The authors reconceptualize adversarial defense as an information-theoretic signal separation problem based on the Multi-view Information Bottleneck (MIB) principle. GUARD aims to simultaneously maximize the preservation of task-relevant information (“informational sufficiency”) and suppress adversarial noise (“purity”). Unlike traditional adversarial training approaches, GUARD introduces no explicit penalty terms and instead allows the information bottleneck to emerge naturally through optimization dynamics. Extensive experiments on five benchmark datasets and nine DMVC models demonstrate that GUARD significantly enhances clustering robustness, restoring performance to near-clean levels while maintaining efficiency—achieving up to 400× faster inference compared to diffusion-based purification methods. The results highlight GUARD’s strong generalizability, transferability, and practicality for robust multi-view clustering in adversarial environments.

**Strengths:**

1. The paper reformulates adversarial defense for multi-view clustering through an information-theoretic lens.
2. GUARD is fully unsupervised and does not rely on model-specific training or labels. This design enables seamless integration with various DMVC models and broad applicability across complete and incomplete multi-view scenarios.
3. Compared to diffusion-based purification methods, GUARD achieves up to a 400× inference speedup, making it highly practical for real-world applications where efficiency matters.

**Weaknesses:**

1. The paper assumes that deep multi-view clustering (DMVC) models are meaningfully threatened by adversarial attacks, but provides no convincing real-world evidence. In unsupervised settings, where no labels exist, the impact of such attacks is ambiguous and less critical than in supervised tasks.
2. The threat model and defense scenario appear largely theoretical. It is unclear why adversarial robustness should be prioritized for clustering tasks that are typically exploratory rather than deployed in high-stakes applications.
3. The proposed information bottleneck perspective is conceptually appealing but lacks rigorous derivation or analytical depth. The method mostly reinterprets existing adversarial purification techniques under new terminology, without offering novel theoretical insights.
4. Experiments are restricted to small, synthetic datasets such as MNIST and FashionMNIST variants, which fail to capture the complexity and diversity of real-world multi-view data. This limits the generalizability and credibility of the empirical results.
5. Considering the unclear real-world significance and the heavy experimental setup, the overall impact of the work may not justify the complexity of the framework. The contribution risks being perceived as a technically competent but conceptually weak exercise.

**Questions:**

Please see weaknesses.

---

### Note · Authors · 2025-11-27

I have read and agree with the venue's withdrawal policy on behalf of myself and my co-authors.